# New In Situ Aerosol Hyperspectral Optical Measurements over 300-700 nm, Part 2: Extinction, Total Absorption, Water- and Methanol-soluble Absorption observed during the KORUS-OC cruise

Carolyn E. Jordan[1,2], Ryan M. Stauffer[3], Brian T. Lamb[4], Michael Novak[3,5], Antonio Mannino[3], Ewan C. Crosbie[2,6], Gregory L. Schuster[2], Richard H. Moore[2], Charles H. Hudgins[2], Kenneth L. Thornhill[2,6], Edward L. Winstead[2,6], Bruce E. Anderson[2], Robert F. Martin[2], Michael A. Shook[2], Luke D. Ziemba[2], Andreas J. Beyersdorf[2,7], Claire E. Robinson[2,6], Chelsea A. Corr[2,8], and Maria A. Tzortziou[3,4]

[1]National Institute of Aerospace, Hampton, Virginia, United States of America
[2]NASA Langley Research Center, Hampton, Virginia, United States of America
[3]NASA Goddard Space Flight Center, Greenbelt, Maryland, United States of America
[4]City University of New York, New York, New York, United States of America
[5]Science Systems and Applications Inc., Lanham, Maryland, United States of America
[6]Science Systems and Applications Inc., Hampton, Virginia, United States of America
[7]California State University, San Bernardino, California, United States of America
[8]Springfield College, Springfield, Massachusetts, United States of America

*Correspondence to*: C. E. Jordan (Carolyn.Jordan@nasa.gov)

**Abstract.** This two-part study explores hyperspectral (300 - 700 nm) aerosol optical measurements obtained from in situ sampling methods employed during the May-June 2016 Korea United States - Ocean Color (KORUS-OC) cruise conducted in concert with the broader air quality campaign (KORUS-AQ). Part 1 focused on the hyperspectral measurement of extinction coefficients ($\sigma_{ext}$) using the recently developed in situ Spectral Aerosol Extinction (SpEx) instrument and showed that 2nd order polynomials provided a better fit to the measured spectra than power law fits. Two dimensional mapping of the 2nd order polynomial coefficients ($a_1, a_2$) was used to explore the information content of the spectra. Part 2 expands on that work by applying a similar analytical approach to filter-based measurements of aerosol hyperspectral total absorption ($\sigma_{abs}$) and soluble absorption from filters extracted either with deionized water ($\sigma_{DI-abs}$) or methanol ($\sigma_{MeOH-abs}$). As was found for $\sigma_{ext}$, 2nd order polynomials provided a better fit to all three absorption spectra sets. Averaging the measured $\sigma_{ext}$ from Part 1 over the filter sampling intervals in this work, hyperspectral single scattering albedo ($\omega$) was calculated. Water-soluble aerosol composition from the DI extracts was used to examine relationships with the various measured optical properties. In particular, both $\sigma_{DI-abs}$(365 nm) and $\sigma_{MeOH-abs}$(365 nm) were found to be best correlated with oxalate ($C_2O_4^{2-}$), but elevated soluble absorption was found from two chemically and optically distinct populations of aerosols. The more photochemically aged aerosols of those two groups exhibited partial spectra (i.e., the longer wavelengths of the spectral range were below detection) while the less-aged aerosol of the other group exhibited complete spectra across the wavelength range. The chromophores of these groups may have derived from different sources and/or atmospheric processes, such that photochemical age may have been only one factor contributing to the differences in the observed spectra. The differences in the spectral properties of these groups was evident in ($a_1, a_2$) maps. The results of the two-dimensional mapping shown in Parts 1 and 2 suggest that this spectral characterization may offer new methods to relate in situ aerosol optical properties to their chemical and microphysical characteristics. However, a key finding of this work is that mathematical functions (whether power laws or 2nd order polynomials) extrapolated from a few wavelengths or a subrange of wavelengths fail to reproduce the measured spectra over the full 300-700 nm wavelength range. Further, the $\sigma_{abs}$ and $\omega$ spectra exhibited distinctive spectral features across the UV and visible wavelength range that simple functions and extrapolations cannot reproduce. These results show that in situ hyperspectral measurements provide valuable new data that can be probed for additional information relating in situ aerosol optical properties to the underlying physicochemical properties of ambient aerosols. It is anticipated that future studies examining in situ aerosol hyperspectral properties will not only improve our ability to use optical data to characterize aerosol physicochemical properties, but that such in situ tools will be needed to validate hyperspectral remote sensors planned for space-based observing platforms.

## 1 Introduction

In Part I (Jordan et al., 2020b) of this two part work, hyperspectral aerosol extinction ($\sigma_{ext}$) measurements from a field campaign in South Korea were used to examine the usefulness of characterizing the spectra with 2nd order polynomial rather than traditional linear fits (i.e., power laws with negative slopes known as Ångström exponents) to the logarithmically transformed spectra. It was found that 2nd order polynomials provided better fits to the data and that using the two-coefficient mapping approach introduced by Schuster et al. (2006) holds promise for providing more detailed information on ambient aerosol size distributions than can be obtained from single Ångström exponents alone. Further, it was demonstrated that extrapolating to wavelengths outside of the measurement range using either fitting approach can lead to large errors in calculated extinctions, suggesting that hyperspectral measurements over broad wavelength ranges are preferable to Ångström exponent based extrapolations. Here, we extend that work to examine the applicability of 2nd order polynomials to represent hyperspectral aerosol absorption measurements and whether

two-coefficient mapping can provide more detailed information on aerosol composition derived from optical measurements compared to that from Ångström exponents.

The data reported in this study are in situ aerosol measurements (Table 1) from a suite of instruments deployed aboard the *R/V Onnuri* during the Korea United States - Ocean Color (KORUS-OC (US-Korean Steering Group, 2015)) cruise around the Korean

peninsula (May 20[th] to June 6[th] in 2016). In conjunction with the airborne air quality mission KORUS-AQ (Al Saadi et al., 2015; Crawford et al., 2020; Tzortziou et al 2018; Thompson et al 2019), these field campaigns were joint South Korean - United States missions to investigate ocean color and air quality within the field of view of the Geostationary Ocean Color Imager (GOCI), that provided both hourly OC and aerosol optical depth (AOD) observations from South Korea's Communication, Ocean, and Meteorological Satellite (COMS). The in situ aerosol measurement suite included both high temporal resolution measurements

(detailed in Part 1) and longer duration filter sampling described in this work. Two sets of filters were collected for post-cruise analyses. The Teflon filters were extracted with water and methanol for subsequent composition and soluble absorption spectra measurements. The glass fiber filters were placed in the center of an integrating sphere to measure total aerosol absorption following the International Ocean Colour Coordinating Group (IOCCG) protocol (IOCCG Protocol Series, 2018).

Spectral measurements of soluble chromophores have become routine for the characterization of brown carbon (BrC) in the ambient atmosphere (e.g., Hecobian et al., 2010; Zhang et al., 2013; Zhang et al., 2017), however, spectral measurements of total aerosol absorption have not. Here, the approach developed by the ocean color community (IOCCG Protocol Series, 2018) has been adopted to assess its applicability to atmospheric aerosols. This is similar to previously reported measurements from particulates filtered from the meltwater of ice and snow (Grenfell et al., 2011) in that ocean (and freshwater) samples are filtered

through glass fiber filters collecting suspended particulates ranging from about 0.7 - 50 µm (Stramski et al., 2015) on the filters. The filter is then placed in the center of an integrating sphere attached to a spectrometer to measure the absorption coefficient ($\sigma_{abs}$) spectra of the particulates. Here, we report hyperspectral measurements from 300-700 nm (in common with the range reported in Part 1 from the in situ Spectral Aerosol Extinction (SpEx) instrument) for $\sigma_{abs}$ and the two sets of soluble (deionized water, DI, and methanol, MeOH) absorption spectra, $\sigma_{DI-abs}$ and $\sigma_{MeOH-abs}$.


Ocean particulates extend to much larger diameters than atmospheric aerosols and exhibit complex absorption spectra with features that include those that arise from specific pigments and physiological state of algae, along with the spectral dependence of non-algal particles such as suspended minerals and black carbon (BC) (e.g., Roesler and Perry, 1995; Stramski et al., 2015; Mouw et al., 2015; IOCCG Protocol Series, 2018). The IOCCG (2018) measurement protocol is known as the quantitative filter technique

(QFT, Mitchell, 1990; Röttgers and Gehnke, 2012; Stramski et al., 2015) because it obtains quantitative $\sigma_{abs}$ via an empirically derived correction factor. As discussed in Grenfell et al. (2011), Stramski et al. (2015), and IOCCG (2018) it is important that the particulates used to derive the correction are representative of the particulates present in the sample population. As will be discussed further in Sect. 2.3, the difference between the size distribution and composition of atmospheric and oceanic particulates suggests that the IOCCG (2018) correction factor may not be applicable to this sample set. However, with a suitably adjusted

correction, this method allows for quantitative measurements of aerosol $\sigma_{abs}$ spectra that can be combined with $\sigma_{ext}$ spectra (as described in Part 1) for the calculation of hyperspectral single scattering albedo ($\omega$).

The ability of aerosols to absorb light depends on their molecular structure, specifically the presence of conjugated (or $\pi$) bonds, heteroatoms (e.g., N or O), and functional group molecular structures (e.g., Jacobson, 1999; Apicella et al., 2004; Andreae and

Gelencsér, 2006; Moosmüller et al., 2009; Chen and Bond, 2010; Desyaterik et al., 2013). The number of such structures determines the wavelength dependence of the absorption spectrum, such that more absorbing molecular structures allow for absorption across a wider range of wavelengths. In the graphitic sheets that comprise BC there are so many $\pi$ bonds, that light is absorbed throughout the ultraviolet (UV) - visible (Vis) - infrared (IR) range (e.g., Yang et al., 2009; Desyaterik et al., 2013; Bond et al., 2013). Organic aerosols, however, contain far fewer such structures such that light is absorbed only over a portion of the

UV-Vis-IR range and may not exhibit absorption in the Vis-IR at all. Non-carbonaceous aerosols, such as dust, are also known to weakly absorb light as a function of their composition, principally according to their proportion of iron oxides (e.g., Sokolik and Toon, 1999: Moosmüller et al., 2012; Shi et al., 2012). Two forms of iron oxide, hematite and goethite, exhibit very different wavelength dependence in their absorption spectra (Schuster et al., 2016).

Given the importance of composition on the absorption of aerosols, it is not uncommon to treat $\sigma_{abs}$ as a function of composition, while treating $\sigma_{scat}$ (and by extension $\sigma_{ext}$, since scattering is the dominant term in extinction) as a function of aerosol size distribution. This separation is not strictly true as mixing state and size distributions have been shown to influence the spectral behavior of $\sigma_{abs}$ (Schuster et al., 2016), while absorption has been shown to influence the spectral behavior of $\sigma_{ext}$ (Eck et al., 2001). Hence, the ability to simultaneously measure hyperspectral $\sigma_{abs}$ and $\sigma_{ext}$ of in situ aerosols along with their chemical

composition and size distributions is anticipated to improve our understanding of the interaction of differing spectral dependencies due to various microphysical aspects of ambient aerosols.

For KORUS-OC, the in situ aerosol instrument suite did not include size distribution measurements or on-line composition, however, there is extensive information about the ambient aerosols observed throughout the KORUS-AQ campaign that can be

used to assess how the ambient aerosol sizes and composition influenced the observed spectral characteristics obtained from the instruments aboard the *R/V Onnuri*. As will be discussed in Sects. 2.1 and 3.2, there were three distinct synoptic meteorological periods during the KORUS-OC cruise (Peterson et al., 2019) that gave rise to aerosol populations that differed in size and composition (Jordan et al., 2020a; Jordan et al., 2020b). These different ambient populations provide the framework that will be used here to assess shifts in the optical properties related to size distribution ($\sigma_{ext}$, following the discussion in Part 1) and

composition ($\sigma_{abs}$, $\sigma_{DI-abs}$, and $\sigma_{MeOH-abs}$).

## 2 Methods

### 2.1 Ship deployment and high-frequency, ambient aerosol measurements

A detailed description of the ship deployment and the high frequency measurements made as part of the in situ aerosol instrument suite deployed on the Korean Institute of Ocean Sciences and Technology (KIOST) *R/V Onnuri* is provided in the companion

paper (Jordan et al., 2020b). In brief, the KORUS-OC cruise of the *R/V Onnuri* sailed first along the east coast of South Korea, then transited to the west (Fig. 1). The in situ aerosol instrument suite measured fine mode aerosols with a 50% cut-off size of 1.3 µm diameter. Filter samples were collected over 3 hour daytime intervals, along with a 12 hour overnight sample, each day from May 22[nd] to June 4[th] (local Korea Standard Time, KST, = UTC + 9) outside of the South Korea's territorial waters (> 12 nautical miles, 22.2 km, from the coast) with mean sampling locations shown in Fig. 1 (see Table S1 for details as some filter samples

overlap in the figure). The locations shown in Fig. 1 are color coded according to the meteorological regimes during the cruise as described by Peterson et al. (2019): Stagnant (May 17[th] - 22[nd], under a persistent anticyclone), Transport/Haze (May 25[th]-31[st],

dynamic meteorology with 4 frontal passages, westerly low-level transport from upwind sources in China, and humid conditions promoting the development of fog and haze under cloudy skies), and Blocking (June 1st - 7th, less persistent stagnation arising from a Rex Block, i.e., adjacent high and low pressure systems with the high poleward of the low) with periods nominally defined from midnight (00:00) through midnight (24:00) local time. Note, in Fig. 1, there is a group of filter samples classified as "transition" collected on May 23rd and 24th as the Stagnant period meteorological conditions gave way to those that defined the Transport/Haze period. Following the discussion in the companion paper, the Transport/Haze filter set is divided into two subsets, those downwind of the Korean peninsula along the east coast and those upwind to the west of the peninsula with the one sample along the south coast assigned to the west group. Details on the specific meteorological conditions and aerosol characteristics will be discussed in Sect. 3.2.

Means for each filter sampling interval (Table S1) were calculated from high temporal resolution data sets described in Jordan et al. (2020b): 3 visible wavelength scattering (450, 532, and 632 nm $\sigma_{scat}$, AirPhoton IN101 nephelometer, Baltimore, MD) and absorption (467, 528, and 652 nm $\sigma_{abs}$, Tricolor Absorption Photometer, TAP, Brechtel, Hayward, CA), and hyperspectral $\sigma_{ext}$ (300-700 nm) from the custom built SpEx instrument. Filter mean $\sigma_{ext}$ calculated at 450, 532, and 632 nm (see Jordan et al., 2020b for details) from the $\sigma_{scat}$ and $\sigma_{abs}$ data is denoted NT $\sigma_{ext}$ here, in contrast to SpEx $\sigma_{ext}$. The high-resolution data set was flagged to identify and remove interceptions of the *R/V Onnuri's* own ship stack emissions (Jordan et al., 2020b). Similarly, here it was also used to assess potential ship plume contamination of the filter set (see the discussion in the supplementary information). Of the 53 pairs of filters collected, no ship plume interceptions occurred for 15. The degree of contamination depended on the duration of the interception as well as on the aerosol property of interest (e.g., Fig. S1), such that in the few instances where the plume contamination (PF1, gray plus symbols) values in scattering or absorption were relatively large their influence over the filter sampling interval was small (compare the solid, All data, and open, PF0, green symbols in Fig. S1). Here, single scattering albedo calculated from IN101 and TAP data provided the most conservative delineation between ambient and ship plume aerosol (Figs. S1 and S2) resulting in 13 of the filter pairs being rejected (Table S2) from most of the analyses presented in Sect. 3 (except as noted). Given the generally small difference in absorption resulting from the ship plume interceptions, all but filter 22 were retained for the assessment of the correction factor for the spectral absorption measurement discussed in Sect. 2.3. Although, filter 22 was rejected from further analyses on the basis of ship plume contamination (Table S2), this particular filter was also so heavily loaded due to polluted ambient conditions that the measured spectrum in the integrating sphere was distorted. Hence, this filter was also excluded from the calculation of the correction factor on the basis of its being overloaded.

## 2.2 Filter preparation and sampling procedures

Two sets of filters were collected for subsequent spectral analyses in the laboratory: Glass fiber filters (Whatman GF/F 47 mm filters 0.7 µm pore size) for total absorption spectra ($\sigma_{abs}$) measured in an integrating sphere attached to a spectrophotometer (see Sect. 2.3 below) and Teflon filters (Fluoropore PTFE 47 mm filters 1 µm pore size) for soluble absorption spectra from deionized water ($\sigma_{DI-abs}$) and methanol ($\sigma_{MeOH-abs}$) extracts from the filters (see Sect. 2.4 below). For each filter set, a blank filter was collected each day during the cruise by briefly putting it in the sampling line without any air flow through the system.

The Teflon filters were taken directly from their package and placed in a filter cassette for sampling. GF/F filters were pre-baked at 450°C for 6 hours in the laboratory prior to the cruise. They were then individually stored in polystyrene petri dishes sealed with Teflon tape, wrapped in aluminum foil, and enclosed in Ziploc bags until placed in a filter cassette for sampling aboard ship.

The two filter cassettes sampled in parallel off the common inlet used for the measurement suite.  The flow rates through the Teflon and GF/F filters were ~24 and ~21 lpm, respectively.  After sampling, filters were individually placed in polystyrene petri dishes, sealed with Teflon tape, wrapped in aluminum foil to limit exposure to light, placed in Ziploc bags, and kept frozen until they were analyzed in the laboratory.

## 2.3  Total absorption coefficient spectra from GF/F samples

Total absorption coefficient ($\sigma_{abs}$) spectra from 300-700 nm at 0.2 nm spectral resolution were obtained from the GF/F set by placing each filter in the center of an integrating sphere (Labsphere DRA-CA-30) attached to a dual beam spectrophotometer (Cary 100 Bio UV-Visible Spectrophotometer).  This methodology is well established in the ocean color community for obtaining particulate $\sigma_{abs}$ spectra from filtered water samples (e.g., Stramski et al., 2015; Neeley et al., 2015; IOCCG Protocol Series, 2018).  The GF/F set were collected during KORUS-OC to test the applicability of this approach to atmospheric aerosol samples.


The absorbance spectra of the GF/F filter set were measured in the laboratory over the course of 4 days with regular air scans (spectra obtained with the filter holder in the integrating sphere without a filter in place) measured to account for any drift in the spectrophotometer measurement.  Absorbance spectra from field blank filters were consistent with laboratory blank filters indicating that handling in the field did not contribute to the measured absorbance of the sample filters.  Blank corrections for the

sample filters were calculated by taking the mean of the air scans and subtracting it from the mean of the field blank spectra.  Multiple scans (2-6) for each filter sample were performed, rotating the filter position between scans to assess variability in the measurement by sampling different parts of the filter with the narrow, collimated beam.  The mean spectrum was then blank corrected.  Two lamps are used in the spectrophotometer to cover the full spectral range.  The tungsten (halogen) lamp provides a stable signal that spans most of the spectral range.  Scanning from longer to shorter wavelengths at 350 nm the instrument switches

to a much noisier deuterium lamp to cover the remainder of the UV range where the intensity of the tungsten light source is too low to make a good measurement.  To minimize the noise present in the UV, a 10 nm boxcar smoothing algorithm was applied to the full wavelength range of the 0.2 nm scans.  The lower limit of detection (LLOD) for the sample data set was calculated from the mean + 3 standard deviations of the field blank filters.  Measurement error for the absorbance scans was estimated to be about 15%.


The dimensionless blank-corrected absorbance (Abs, also known as the spectral optical density of the filter (Stramski et al., 2015)) is then used to calculate the $\sigma_{abs}$ via

$$\sigma_{abs}(\lambda) = \frac{LN(10) \times Abs(\lambda) \times f_a}{V_{air} \times \beta} \tag{1}$$

where $f_a$ is the area of the filter (in $m^2$), $V_{air}$ is the volume of air sampled through that filter area (in $m^3$), $\beta$ is a term used to correct

for pathlength amplification of the signal by the filter and collected particles (Butler, 1962; Kiefer and Soo Hoo, 1982; Kishino et al., 1985), and the LN(10) term converts the base 10 logarithm absorbance measurement (= $-\log_{10}$ T, where T is transmittance) to a natural logarithm as used for ambient environmental observations.  These terms result in units of $m^{-1}$ for $\sigma_{abs}(\lambda)$, which are inconvenient for atmospheric data.  Hence, an additional factor of $10^6$ is also used to convert the units to $Mm^{-1}$ for the data set herein.  The $\beta$ correction accounts for the scattering enhancement by the fibers within the filter itself, as well as by the particles

collected on and within the filter medium and is defined as the ratio of the measured filter absorbance to the true absorbance of the sample.

β has been determined empirically with the standard protocol (IOCCG Protocol Series, 2018) adopting the parameterization determined by Stramski et al. (2015), denoted here as $\beta_s$,

$$\beta_S = 3.096 \times Abs^{-0.0867} \tag{2}$$

The ocean optics and biogeochemistry protocol (IOCCG Protocol Series, 2018) notes that "understanding ... the types of particle assemblages for which the formulated pathlength amplification correction is representative" is important. Here, those assemblages included nearshore mineral-dominated and red tide samples, mixtures of phytoplankton species, and other coastal and offshore particle assemblages spanning a size range of about 0.7 to 50 µm (Stramski et al., 2015). A similar study that used a different instrument configuration to measure the absorption of particles filtered from snow and ice meltwater (Grenfell et al., 2011) noted that the calibration standard used for soot was carefully filtered to remove particles that were too large to be representative of the ambient samples. Figure S3 shows the relationship between $\sigma_{abs}$ calculated using Eqs. (1) and (2) from the filter measurements in the integrating sphere compared to $\sigma_{abs}$ measured by TAP (values at 467, 528, and 652 nm wavelengths are all plotted together here). Using no correction at all ($\beta = 1$) results in values much higher than the TAP values (with a slope of 2.361), while the $\beta_s$ correction leads to values that are much lower (slope of 0.593).

Fresh soot particles from diesel engines are small, typically on the order of tens of nm (Bond et al., 2013), hence the size range measured by the in situ aerosol instrument suite aboard *R/V Onnuri* ($\leq 1.3$ µm diameter) likely included smaller particles (as well as particles with a very different composition) than those used to derive $\beta_s$. Given that TAP is a commercial instrument with a well characterized correction (Ogren et al., 2017) suitable for atmospheric BC aerosols, along with the known importance of using a $\beta$ correction determined from particles representative of the sample data, we have greater confidence in the magnitude of the TAP values than the $\beta_s$ correction for this data set. Empirically determining an appropriate $\beta$ for the samples obtained in this study was beyond the scope of this project. However, taking advantage of the TAP data set, $\beta$ for the KORUS-OC data ($\beta_{K-OC}$) was obtained by scaling $\beta_s$ to fit the TAP data ($\beta_{K-OC} = 0.593 \times \beta_s = 1.836 \times Abs^{-0.0867}$, see Fig. S3) resulting in quantitative $\sigma_{abs}$ spectra. Error propagation (i.e., the propagation of the uncertainty of each term in (Eq. 1)) was used to estimate the uncertainty in the $\sigma_{abs}$ spectra (Eq. (1)) from the estimated 15% error of the absorbance spectra, the standard deviation of the slope ($\pm 0.034$) used to estimate $\beta_{K-OC}$, the uncertainty in the filter area (based on $\pm 1$ mm in diameter of the estimated filter sampling diameter of 42.6 mm of the 47 mm diameter filters) and the standard deviation of the mean volume of air sampled for each filter calculated from 1 s flow rate data. Over 300-700 nm the estimated error in $\sigma_{abs}$ is ~17% (Table 1).

The spectrometers used to measure $\sigma_{ext}$ and $\sigma_{abs}$ had different spectral resolutions. Hence, in order to calculate single scattering albedo ($\omega$) these spectra sets were averaged into 2 nm bins over 300-700 nm. Then spectral $\omega$ was calculated via

$$\omega(\lambda) = \frac{\sigma_{ext}(\lambda) - \sigma_{abs}(\lambda)}{\sigma_{ext}(\lambda)} \tag{3}$$

## 2.4 Soluble absorption coefficient spectra from Teflon samples

The Teflon filters were cut in half in order to extract water-soluble aerosol components from one half that could be further analyzed via ion chromatography and aerosol mass spectrometry (see Sect. 2.5), and non-water-soluble components from the other half using an organic solvent. Here, methanol (MeOH) was used for the organic solvent as both water and MeOH solutions were

compatible with the waveguide used to measure the absorbance of the dissolved aerosol components in the extracts. Half of the filter was placed in a clean 15 ml polypropylene centrifuge tube (Corning 430052; triple rinsed, soaked overnight, and triple rinsed again with Milli-Q 18 MΩ deionized (DI) water) and extracted in 10 ml of DI water by hand shaking the tube for 60 s. Tests were performed that showed this method extracted the soluble chromophores equally well as with 60 min of sonication, hence the faster approach was employed. Extracts were filtered to remove any insoluble particles using polypropylene Soft-Ject disposable 12 ml syringes (Henke Sass Wolf) and 0.2 μm pore size PTFE-membrane filters (Cole-Parmer).

The other half of the filter was placed in a clean 15 ml glass vial with a Teflon cap (rinsed with spectrophotometric grade ($\geq$ 99.9%) MeOH (Sigma Aldrich product number 154903) and dried in a fume hood) and extracted in 10 ml of MeOH by hand shaking the vial for 60 s. Extracts were filtered to remove particles using a glass syringe with a 0.2 μm pore size PTFE-membrane filter (Cole-Parmer). Unfortunately, some of the MeOH extracts vials were contaminated during handling and are marked as "missing" in Table S1. As a result, there are fewer MeOH-soluble absorption spectra in the data set than DI-soluble absorption spectra. In all cases where there are both spectra, greater absorption was observed from the MeOH extracts.

A liquid waveguide capillary cell (LWCC; World Precision Instruments LWCC-3100, 100 cm pathlength) attached to an SM240 spectrometer (Spectral Products, Putnam, CT, ~0.4 nm spectral resolution) with a DH-2000-BAL light source (Ocean Optics, Dunedin, FL) operated with a 2 s integration time was used to measure the absorbance of the soluble aerosol chromophores in each filter extract. Note, absorbance (Abs) is the measured quantity, it is not the same as the absorption coefficient ($\sigma_{sol-abs}$, where the notation sol-abs is used to distinguish soluble from total absorption coefficients). $\sigma_{sol-abs}$, the relevant atmospheric quantity, is derived from Abs. The relationship between these values is explained below. Each set of measurements in the laboratory started with a dark count spectrum followed by an alternating sequence of a reference spectrum from the solvent alone, then a sample spectrum from a filter extract, then another reference spectrum, etc. Periodically, the LWCC would be cleaned with 0.5 N HCl and acetone (for HPLC, $\geq$ 99.9%, Sigma Aldrich product number 270725).

The Abs of the dissolved materials in the extract measured in the LWCC was calculated via

$$\text{Abs}(\lambda) = -\log_{10} T(\lambda) = -\log_{10}\left(\frac{I_S(\lambda)}{I_0(\lambda)}\right) \tag{4}$$

where, T is the transmittance, i.e., the fraction of the total light ($I_0$) that passes through the sample ($I_S$). To account for any drift in the light intensity from the lamp, the reference intensity ($I_0$) before ($I_{01}$) and after ($I_{02}$) each sample ($I_S$) was averaged and used as the reference for that sample. Correcting each term for the dark counts of the spectrometer ($I_D$), Eq. (4) can then be expressed as (wavelength dependence not shown for clarity)

$$Abs = -\log_{10}\left(\frac{(I_S - I_D)}{((I_{01} - I_D) + (I_{02} - I_D))/2}\right) \tag{5}$$

For each extract the intensity spectra were acquired for 40 - 60 s. The mean and its standard deviation for each $I_D$, $I_{01}$, $I_{02}$, and $I_S$ were calculated. The uncertainty for each absorbance spectrum was obtained using error propagation from the standard deviations of the intensity spectra. Extracts from field blank filters were also measured and used to calculate the lower limit of detection (LLOD) in absorbance (using the mean + 3 standard deviations of the blanks). The LLOD is wavelength-dependent and was found to be higher for the MeOH set than the DI set. In contrast, the upper limit of detection (ULOD) was constant and found to be 0.8 Abs for both solvents.

$\sigma_{sol-abs}$ for the soluble aerosol chromophores in the extracts were then calculated from the blank-corrected Abs spectra via

$$\sigma_{sol-abs}(\lambda) = Abs(\lambda) \bullet \frac{V_s}{V_a \bullet l} \bullet LN(10) \qquad (6)$$

where $V_s$ is the volume of solvent used to extract the filter (in liters), $V_a$ is the volume of air sampled by the filter (in liters), l is the absorbing path length of the capillary cell (in meters), and LN(10) converts the base 10 log used for absorbance to natural log typically used for atmospheric quantities (Hecobian et al., 2010). Again, the units were converted from $m^{-1}$ to $Mm^{-1}$ per the convention commonly used for atmospheric data sets. $\sigma_{sol-abs}$ spectra are reported for the wavelength range of 300-700 nm with problematic wavelengths (due to saturation of the detector) removed resulting in evident gaps in the spectra (e.g., Fig. 2). The uncertainty in $\sigma_{sol-abs}(\lambda)$ was assessed using error propagation and found to be about ± 30%, principally due to the uncertainty in the volume of the solvent used to extract the filters.

Previous authors (e.g. Hecobian et al., 2010; Zhang et al., 2013) have calculated $\sigma_{sol-abs}(\lambda)$ at 365 nm ($\sigma_{sol-abs}$(365nm)) by averaging over the 360-370 nm range of the spectrum in order to use it as a single value proxy for soluble organic aerosol chromophores. This avoids contributions from inorganic nitrate absorption that occurs at wavelengths < 330 nm. We adopt this proxy for the discussion herein.

## 2.5 Chemical analyses of water-soluble extracts from Teflon filters

In parallel with analysis in the LWCC, the water-soluble extracts were analyzed for their chemical composition using both a Dionex ICS-3000 Ion Chromatography (IC) System (Thermo Fisher Scientific, Waltham, MA) and an Aerosol Mass Spectrometer (AMS, specifically an HR-ToF-AMS, Aerodyne Research, Inc., Billerica, MA). Lower limits of detection for the IC data set were calculated from the mean plus 3 times its standard deviation of the field blanks. Most of the below detection IC data were among the sea salt and dust ions ($Na^+$, $Cl^-$, $Mg^{2+}$, and $Ca^{2+}$), with a few occurrences found among $NO_3^-$, $K^+$, and $C_2O_4^{2-}$. No samples were below detection in $NH_4^+$ and $SO_4^{2-}$. These results are consistent with the expectation that anthropogenic pollution sources dominate the fine fraction of aerosols, the population sampled by this measurement suite with a 50% size cut of 1.3 µm diameter (Jordan et al., 2020b). No sea salt corrections were applied to any of the reported concentrations.

The water-soluble extract was aerosolized using a nebulizer supplied with particle-free air and subsequently dried followed by sampling with the AMS. The mass spectrum measured by the AMS reflects the composition of the low volatility (e.g. particulate) but non-refractory, water-soluble fraction of components collected on the filter. Unlike the IC data, it is not possible to quantify the atmospheric mass concentrations of the aerosol components measured here with the AMS. The nebulizers used to disperse the DI extracts into aerosols do so with an unknown liquid flow rate such that the original atmospheric aerosol mass concentrations in ambient air cannot be calculated. Hence, in order to make comparisons across the data set, the AMS data is presented in terms of ratios, either the ratio of the major chemical groups (sulfate, ammonium, nitrate, chloride, and organic) to their summed total mass (Total) or the ratio of individual m/z groups to the total organic mass (Organics). Note, since organic sulfate and nitrate compounds can contribute to the sulfate and nitrate measured by AMS the notation used for ions is not used here. Throughout this paper the overlapping IC and AMS measurements are distinguished by using the ionic notation for the former (i.e., $SO_4^{2-}$ and $NO_3^-$) and the terms (i.e., sulfate and nitrate) for the latter.

The summation over all m/z ≥ 12 used to calculate Organics included negative values that arise from the subtraction of a reference spectrum (filtered airflow to remove the particles) from the unfiltered airflow containing aerosols in the AMS. Typically, the negative values were a minor contribution. However, in some cases they were large. Hence, the fractional contributions of each m/z to the sum (denoted here as "f_m/z") were normalized across only positive values. The normalized value was used to approximate above detection contributions for each m/z to Organics. Note, in the assessment of individual contributions to Organics the range of values examined was truncated to the m/z 12-73 range. For all but 6 filters this captured ≥ 0.9 of Organics.

## 3  Results and Discussion

### 3.1  Filter spectra set overview

Each panel of Fig. 2 shows one filter from each meteorological period as shown in Fig. 1 to illustrate differences observed across the entire data set. For the measured filter spectra ($\sigma_{abs}$, black, $\sigma_{MeOH-abs}$, purple, and $\sigma_{DI-abs}$, blue, curves Fig. 2) shading is used to indicate estimated measurement errors in absorption coefficients (see Table 1) as described in Sect. 2. The larger errors for the soluble absorbers make this shading easier to discern than that for $\sigma_{abs}$ on the log scale needed here to show all of the spectra together. For the higher temporal resolution measurements averaged over the filter sampling periods shading (SpEx $\sigma_{ext}$, red curves) and error bars (NT $\sigma_{ext}$, red pluses, and TAP $\sigma_{abs}$, gray pluses) indicate one standard deviation of the mean. Hence, the standard deviations reflect the variability of the ambient atmosphere over each filter sampling period. For three of the examples, little ambient variability in $\sigma_{ext}$ was observed, while the example from the Blocking period indicates rapidly changing conditions that occurred during the 12 h overnight sample from June 1st to 2nd (see Part 1).

Key features to note in Fig. 2 include the spectral variability in $\sigma_{abs}$, where sometimes it is smoothly varying (e.g., Blocking) and sometimes there are spectral features evident (e.g., the UV portion of the Stagnant spectrum). The spectral features (i.e., enhanced absorption over a limited wavelength range compared to the rest of the spectrum) likely arise from specific molecular structures within the ambient aerosols that absorb light over a limited wavelength range as discussed in the introduction. Also note the changing relationship between $\sigma_{MeOH-abs}$ and $\sigma_{DI-abs}$, where they can be very similar across the spectrum (e.g., Transport/Haze - East) or diverge substantially at longer wavelengths (e.g., Blocking). Finally, note the partial spectra for $\sigma_{MeOH-abs}$ and $\sigma_{DI-abs}$, where only a portion of the spectrum is above detection. These partial spectra vary considerably in the above detection wavelength range with the Transport/Haze - West example showing an extreme case where all of the $\sigma_{MeOH-abs}$ is above detection, while little of the $\sigma_{DI-abs}$ is above detection. In contrast, for the Stagnant example both of these spectra are partial, but span most of the wavelength range. The differences between the spectra obtained from the DI and MeOH extracts for any given sample arises from differing solubilities of the chemical components of the aerosols in those two solvents. Differences in the soluble spectra across the sample set arise from differences in the ambient aerosol population across the meteorological regimes of the campaign.

DI extracts were analyzed for composition enabling analyses to relate composition to optical properties as will be discussed in Section 3.2. Also in that section, previously published work from the KORUS-AQ campaign is used to provide greater context for the differences in the aerosol populations sampled during the three meteorological regimes that occurred during the cruise. In Section 3.3 fits to the spectra set from 2nd order polynomials are compared to those from power laws. This section also includes a preliminary examination of how the different aerosol populations map into the two parameter space described by the 2nd order polynomial coefficients providing a potential avenue to relate aerosol physicochemical properties to spectral optical characteristics.

In Section 3.4, single scattering albedo ($\omega$) spectra calculated from $\sigma_{abs}$ and $\sigma_{ext}$ are used to illustrate the spectral features in $\sigma_{abs}$ that more clearly manifest in $\omega$. The in situ aerosol instrument suite aboard the *R/V Onnuri* did not include on-line aerosol composition measurements that could have been used to interpret the absorption features in the $\sigma_{abs}$ spectra limiting chemical attribution to the various features observed. However, the examples provided show that future studies that combine hyperspectral total absorption with composition measurements that capture the total aerosol (not just the water-soluble components) may enable such linkages to be made.

## 3.2 Water-soluble aerosol composition and 365 nm optical properties compared across meteorological regimes during KORUS-OC

The synoptic meteorology of the KORUS-AQ campaign described by Peterson et al. (2019) led to differing aerosol populations at the surface across South Korea (Jordan et al., 2020a) both in terms of composition and size distributions of fine mode aerosols (i.e., particulate matter with diameters $\leq 2.5$ $\mu$m, PM2.5). PM2.5 was most elevated during the Transport/Haze period, followed by the Stagnant period (Peterson et al., 2019). Stagnant conditions fostered the accumulation of locally photochemically produced secondary organic aerosol (SOA) under clear skies (Kim et al., 2018; Nault et al., 2018; Choi et al., 2019; Peterson et al., 2019; Jordan et al., 2020a). During the Transport/Haze period transport from the west/northwest brought polluted air from China to S. Korea under cloudy humid conditions that promoted rapid local heterogeneous production of secondary inorganic aerosols (SIA) within a shallow boundary layer producing the highest PM2.5 concentrations during KORUS-AQ (Peterson et al., 2019; Eck et al., 2020; Jordan et al., 2020a). During this period PM2.5 was much greater than PM1 (i.e. aerosols with diameters $\leq 1$ $\mu$m) at Olympic Park in Seoul, often by 20 - 40 $\mu$g/m$^3$. In contrast, at the end of the Stagnant period PM2.5 only exceeded PM1 by margins $\leq 10$ $\mu$g/m$^3$ (Jordan et al., 2020a). Hence, the size distributions as well as concentrations and compositions differed between these periods. The Blocking period (limited transport, occasional brief stagnant periods, and rain to the south (Peterson et al., 2019)) began following a frontal passage that brought clean air from the north. As discussed in the companion paper (Jordan et al., 2020b), the fine mode aerosols sampled aboard *R/V Onnuri* during this period were likely due to small absorbing aerosols from ship emissions in the West Sea.

The aerosol composition measured via the AMS from the nebulized water-soluble filter extracts (Fig. 3) are consistent with the previously reported aerosol composition (Kim et al., 2018; Nault et al., 2018, Choi et al., 2019; Peterson et al., 2019; Jordan et al. 2020a). Dramatic shifts in composition across the differing meteorological regimes are evident in Fig. 3 with organics contributing 85% to the total AMS mass during the Stagnant period versus 33% and 45% when the ship was downwind (May 25[th] - 27[th]) and upwind (29[th] - 31[st]), respectively, of the peninsula during the Transport/Haze period. There is also a clear shift in the water-soluble organic composition between the downwind and upwind sides of the peninsula with m/z 44 dominant downwind and m/z 43 contributing a larger fraction to the organic mass on the upwind side (Fig. 3). m/z 44 is predominantly attributed to the $CO_2^+$ fragment originating from carboxylic acids, to which oxalate is a contributor, and is the basis for positive matrix factorization (PMF) techniques to quantify secondary organic aerosols (SOA; Zhang et al. 2005) and highly oxidized organic aerosol (i.e., OOA-1 from Ulbrich et al., 2009). m/z 43 results from both oxidized and aliphatic fragments, $C_2H_3O^+$ and $C_3H_7^+$ respectively, and is enhanced relative to m/z 44 for less-oxygenated organic aerosol (e.g., OOA-2 from Ulbrich et al. (2009)). Likewise, higher ratios of m/z 44:43 are indicative of lower volatility, more photochemically-aged OOA while lower ratios are linked to semi-volatile oxygenated compounds (Ng et al., 2010).

The upwind (Transport/Haze - West) vs. downwind (Transport/Haze - East) shift is also evident in the IC analyses of these DI extracts (Fig. 4). $NO_3^-$ was up to an order of magnitude greater downwind than upwind during Transport/Haze, while peak $SO_4^{2-}$ nearly doubled, consistent with previous reports of local South Korean heterogeneous secondary production of these constituents (Jordan et al., 2020a). Throughout the cruise, $SO_4^{2-}$ was generally fully neutralized by $NH_4^+$, i.e. in the form $(NH_4)_2SO_4$ (Fig. 4). $K^+$ was linearly related to $NH_4^+$ throughout the cruise with an $r^2 = 0.78$ (Fig. 4). In contrast, $C_2O_4^{2-}$ measured by IC exhibited elevated values both for the Stagnant (and transition) and downwind Transport/Haze samples (Fig. 4). Organic aerosol was elevated during the Transport/Haze period (Jordan et al., 2020a), but it did not increase as rapidly or dramatically as SIA, hence the decreased percent contribution to the total aerosol mass in the AMS samples (Fig. 3). During the Blocking period organics again dominated the composition (Fig. 3) but not to the extent observed during the Stagnant period (64% vs. 85%). For all aerosol components (organics and inorganics) concentrations during this period tended to be relatively low (Fig. 4).

The measured optical properties exhibited differing relationships to the water-soluble aerosol chemical components as illustrated in Fig. 5. For example, $\sigma_{ext}$(365 nm) was well correlated with the mass concentrations of $NH_4^+$ and $K^+$, but poorly correlated with $C_2O_4^{2-}$ (0.76, 0.72, and 0.16, respectively, Fig. 5). The soluble absorption (both DI and MeOH, but only DI shown) was well correlated with $C_2O_4^{2-}$, but poorly correlated $NH_4^+$ and $K^+$ (e.g., $\sigma_{DI-abs}$(365 nm) $r^2 = 0.67$, 0.22, and 0.25, respectively, Fig. 5). This is not unexpected as extinction is related to aerosol mass loading and size distributions, while soluble absorption is a function of soluble aerosol composition. Total aerosol absorption was not well correlated with any of the water-soluble aerosol components (as expected, since BC dominates total absorption in the atmosphere and is not soluble). However, as shown in the companion paper, peak $\sigma_{abs}$ tended to coincide with peak $\sigma_{ext}$ leading to moderate $r^2$ values of ~0.5 depending on wavelength (Fig. 5). This is likely a reflection of mass loading, such that when pollutant aerosol concentrations downwind of South Korea were high, both scatterers and BC were elevated in the air mass. Single scattering albedo clarifies an interesting feature evident in $\sigma_{abs}$, i.e., the total absorption is sensitive to organic absorbers at 365 nm (i.e., BrC). There is little difference in $\omega$(365 nm) and $\omega$(532 nm) for the Transport/Haze and Blocking periods, however, there is a substantial reduction in $\omega$(365 nm) for the Stagnant (and transition) period samples (Fig. 5).

This is a striking observation due to the fact that the Stagnant period aerosols were dominated by local photochemically produced SOA. $K^+$ mass concentrations were relatively low during this period, while elevated $C_2O_4^{2-}$ concentrations at this time were independent of $K^+$ (Fig. 4). Both $K^+$ and $C_2O_4^{2-}$ have known biomass burning sources and have been used as tracers for biomass burning (e.g., Park et al., 2013; Park and Yu, 2016; Szép et al., 2018; Johansen, et al., 2019). However, they also have other sources that can complicate their usage for such a purpose. For example, $C_2O_4^{2-}$ can be formed by aqueous-phase cloud chemical processes (e.g., Huang et al., 2006; Ervens et al., 2011) and fine fraction $K^+$ can arise from fertilizer use (e.g., Szép et al., 2018; Han et al., 2019), fireworks (Hao et al., 2018; ten Brink, et al., 2019), and anthropogenic combustion sources (Jayarathne et al. 2018; Han et al., 2019; Yan et al., 2020) such as coal, garbage incineration, and cooking fuels, with charcoal in particular exhibiting enhanced $K^+$ (Jayarathne et al. 2018). Levoglucosan has also been used as a biomass burning tracer, however, its relative contribution has been shown to decrease with smoke plume age (Cubision et al., 2011) and recent work indicates that it is short-lived in the atmosphere such that it is useful for fresh emissions, but not for aged biomass burning aerosol (Wong et al., 2019). Figure 3 shows m/z 60 from the nebulized water-soluble extracts was negligible throughout the campaign. m/z 60 is typically attributed to the $C_2H_4O_2^+$ fragment from levoglucosan (Schneider et al, 2006) and is enhanced for the biomass burning organic aerosol PMF factor (Cubison et al., 2011). All of these data suggest that during the Stagnant period photochemically produced SOA contributed to absorption at 365 nm without any evident indication of biomass burning. However, the Wong et al. study also

reported that absorbing high molecular weight biomass burning aerosol is relatively stable and much longer lived in the atmosphere than low molecular weight biomass burning aerosol components such as levoglucosan. The implications of this will be discussed further in Sect. 3.3.


The difference between $\omega$(365 nm) and $\omega$(532 nm) for the absorbing aerosols of the Stagnant period and Transport/Haze period show that the elevated $C_2O_4^{2-}$ and elevated $\sigma_{DI\text{-}abs}$ do not fully capture differences across these populations of absorbing organic aerosols. That is, decreases in $\omega$ from 532 to 365 nm are small for the Transport/Haze population, in contrast to the large decrease for the Stagnant (and transition) aerosols. The molecular structures that absorb light differ between the two groups such that the

photochemistry of the Stagnant period appears to have produced molecularly different SOA than was produced under the cloudy hazy conditions of the Transport/Haze period.

It must be emphasized that the chemical contributions discussed in this section only include aerosol components that were soluble in water and hence, only represent a portion of the ambient chemical composition influencing the $\sigma_{ext}$, $\sigma_{abs}$, and $\sigma_{MeOH\text{-}abs}$

measurements. $\sigma_{DI\text{-}abs}$ contributed only a fraction to $\sigma_{abs}$, and $\sigma_{abs}$ were typically an order of magnitude smaller than $\sigma_{ext}$. Hence, relationships that may be explored within this data set between chemical composition and optical properties are constrained by these limitations. Future work would benefit from concurrent in situ aerosol AMS sampling, as well as size distribution measurements, to more fully relate aerosol composition to optical properties. Even so, chromophores may constitute only a small fraction of the aerosol mass (e.g., Bones et al., 2010). Hence, changes in $\sigma_{abs}$ may or may not be tightly coupled to aerosol

composition measurements. The development of in situ hyperspectral techniques to measure the optical properties of in situ aerosols, along with the chemical and microphysical properties of those aerosols is expected to reduce some of the uncertainties in assessing the relationships among the full suite of ambient aerosol characteristics.

### 3.3 Characterizing spectral shapes with 2nd order polynomial coefficients

While isolating individual wavelengths from the spectral data can be informative as illustrated in Sect. 3.2, the anticipated value

of these hyperspectral measurements comes from examining the full spectra. In Part 1 (Jordan et al., 2020b) it was shown that the logarithmically transformed extinction spectra were better fit with a 2nd order polynomial than a line, due to curvature of the spectra in log space. Similarly, for the filter spectra sets 2nd order polynomials,

$$LN(p(\lambda)) \ = \ a_0 + a_1 LN(\lambda) + a_2 (LN(\lambda))^2 \qquad\qquad (7)$$

yielded better fits to the measured spectra than linear fits (i.e. representing a power law),

$$LN(p(\lambda)) \ = \ a + bLN(\lambda) \qquad\qquad (8)$$

of the spectra as shown by the improvement in the fit residuals (Fig. 6). An example set of spectra (Fig. S4) is provided in the SI to illustrate the difference in the residuals from these two fits. Note, that all three absorption spectra tended to exhibit positive curvature in contrast to extinction spectra that are all negative for the filter mean spectra set (e.g., Fig. S4). In Part 1, it was shown that the expanded wavelength range in $\sigma_{ext}$ led to smaller values of $\alpha_{ext}$ due to the negative curvature of the spectra. Here, the

positive curvature of the absorption spectra leads to the opposite result, i.e., larger values of $\alpha_{abs}$ compared to those shown in Part 1 calculated from the visible TAP wavelengths. Hence, when comparing Ångström exponents across data sets the role of curvature requires that the wavelength range used to calculate $\alpha$ be taken into account. The values may not be directly comparable as will be discussed further below.

The residuals from the 2nd order polynomial and linear fits to each spectra set were binned into 20 bins spanning the full range of differences between each measured spectrum and fit spectrum (Fig. 6). The range of differences is smaller leading to narrower bins for the 2nd order polynomial fits than the linear fits. The normal distribution around zero of the 2nd order polynomial fits shows they provide a better fit to the data. Following the approach used in the companion paper, the additional information that may be gained from the coefficients of the 2nd order fits ($a_1$ and $a_2$, Eq. (7)) than from the Ångström exponent, $\alpha$ (= -b, Eq. (8))

provided by linear fits is examined here.

As discussed in detail in Part 1, the coefficients between the two fits are related to each other by the derivatives of Eqs. (7) and (8), where,

$$\alpha = -\frac{d(LN(p(\lambda)))}{dLN(\lambda)} = -b = -(a_1 + 2a_2(LN(\lambda_{ch}))) \tag{9}$$

Here, the notation $\lambda_{ch}$ is used to denote that there is one wavelength for which $\alpha$ and ($a_1$,$a_2$) yield equivalent results for any given spectrum. $-2LN(\lambda_{ch})$ can be thought of as the slope of a line that describes that relationship in ($a_1$,$a_2$) space. This is illustrated in Fig. S5, where for each spectra set $\lambda_{ch}$ was calculated from the pair of fits to each spectrum. In the case of $\sigma_{abs}$, only a very narrow range of values was found such that $\lambda_{ch} = 0.47$. Using this value in Eq. (9), one can map the set of parallel $\alpha$ lines into ($a_1$,$a_2$) space using the range of values obtained for these coefficients from the two sets of fits. For any given $\alpha$, spectra with differing curvature

will map into different pairs of ($a_1$,$a_2$) along that line (Fig. S5). A plot of the individual $\alpha$ values obtained from the $\sigma_{abs}$ set (Fig. 7) shows those points are aligned according to the map in Fig. S5. Note, the color bars for $\alpha$ used in Fig. 7 match those in Fig. S5 (right panels). For $\sigma_{abs}$, the range of $\alpha$ is 1.08 to 2.80. These are larger values than the range shown in Part 1 from the visible wavelength TAP data where $\alpha_{abs} \sim 0.5 - 1$ (Jordan et al., 2020b). As noted above, positive curvature and a wavelength range extending further into the UV leads to the higher $\alpha_{abs}$ values here. Previous studies have used $\alpha_{abs}$ to distinguish absorbing aerosol

composition (e.g., $\alpha_{abs} = 1$, fresh urban-industrial BC, $\geq 2$, desert dust, with intermediate values indicative of biomass burning aerosols (Russell et al., 2010)). However, Schuster et al. (2016) caution that mixing state and size distribution play a role in $\alpha_{abs}$ along with composition.

The ($a_1$,$a_2$) mapping can be used to examine the distribution of samples according to the meteorological regime (Fig. 8) as described

in Sect. 3.2. It is interesting to note that for the Stagnant period $\alpha_{abs} \geq 2$, suggesting desert dust, but that is not consistent with what is known about the ambient aerosol population that was dominated by fine mode SOA at the time. As discussed in Sect. 3.2, the high ratio of m/z 44:43 in the DI extracts for these samples suggest these aerosols were more photochemically aged OOA with lower volatility organic components (Ng et al., 2010). This is a somewhat confounding observation given the significant production of SOA during this period. It is possible that non-BC contributions to $\sigma_{abs}$ were decoupled from the composition of the bulk of the

SOA mass concentrations as has been suggested by Bones et al. (2010). Recall, that Wong et al. (2019) report high molecular weight absorbing components in biomass burning aerosol are relatively stable in the atmosphere. Early in the Stagnant period (May 18th, 4 days prior to the Stagnant samples measured aboard the *R/V Onnuri*) smoke from Siberian wildfires was transported into the study region (Peterson et al., 2019). With limited transport and no rain, it is possible that high molecular weight absorbing components in the aged Siberian biomass burning aerosols persisted in the region until the end of the Stagnant period contributing

to elevated $\alpha_{abs}$ from what would be expected from relatively fresh urban BC emissions accumulating in a Stagnant air mass.

Another possibility could be photochemically produced chromophores among the SOA. Or as Schuster et al. (2016) suggest factors related to the mixing state may be important here.

While $\alpha_{abs}$ values were lower, closer to expectations of BC dominated absorbing aerosol downwind of the peninsula during Transport/Haze (Figs. 7 and 8), there was not clear separation in fit coefficient values across the meteorological periods. The separation observed in the extinction coefficients (Figs. 7 and 8) is less distinct than that of the hourly mean $\sigma_{ext}$ spectra fits reported in the companion paper. This may be due to the longer sampling intervals of the filter spectra set, which may also be a contributing factor to less distinct separation in fit coefficients to $\sigma_{abs}$ across meteorological regimes (Fig. 8). Note, the mapping of $\alpha_{ext}$ isn't strictly parallel in $(a_1, a_2)$ space as it was for $\alpha_{abs}$, due to a wider range in $\lambda_{ch}$ values (= 0.41 - 0.47, Fig. S5) that results in a fanned out spread of $\alpha_{ext}$ (Figs. S5 and 7).

A similar separation in the distribution of $\alpha$ for the soluble absorbers leads to what appear to be 2 distinct branches in $(a_1, a_2)$ space with lower $\alpha$ values clustered over a narrower range of $(a_1, a_2)$ values and larger $\alpha$ values spanning a broad range in $(a_1, a_2)$ extending to both positive and negative extremes in the observed $a_2$ range (Fig. 7). This mapping arises from the relatively broad range of $\lambda_{ch}$ = 0.34 - 0.47 for both $\sigma_{DI-abs}$ and $\sigma_{MeOH-abs}$ (Fig. S5). The differing ranges of $\lambda_{ch}$ across the spectra sets is due to the presence (or absence) of partial spectra, i.e., spectra for which longer wavelengths are below detection and hence, not included in the fit. The fit coefficients are sensitive to the wavelength range of the fit, such that shorter spectra can exhibit more extreme values over the $(a_1, a_2)$ range. Further, the limited wavelength range shifts the value of $\lambda_{ch}$ to shorter wavelengths. For the individual measured $\sigma_{ext}$ spectra set reported in the companion paper $\lambda_{ch}$ ranged from 0.36 - 0.46. Mean $\sigma_{ext}$ spectra over the filter sampling intervals narrowed and shifted that range to slightly longer wavelengths, 0.41 - 0.47 as noted above. There were no partial spectra for $\sigma_{abs}$ leading to $\lambda_{ch}$ = 0.47, while for the soluble absorbers there were many partial spectra as illustrated in Fig. 9. Here, all of the above detection $\sigma_{DI-abs}$ at 0.315 μm are shown, with below detection values at each increment in wavelength across the spectrum absent in subsequent panels (Fig. 9). The magnitude of the soluble absorption coefficients were not separated in $(a_1, a_2)$ space (Fig. 9), however, the $\alpha$ values and meteorological regimes (Figs. 7 and 8) exhibit clear separation. These results are consistent with expectations given the spectral dependence of absorption as a function of the number of conjugated bonds, heteroatoms, and functional groups present in the organic component of aerosols (see Sect. 1). Hence, it is not unexpected to find a variable range of wavelengths over which above detection soluble absorption is found for ambient particles. $(a_1, a_2)$ mapping may therefore be useful for relating optical properties to chromophores.

All of the Stagnant $\sigma_{DI-abs}$ and $\sigma_{MeOH-abs}$ spectra were partial, while none of the Transport/Haze - East were partial leading to separation in $(a_1, a_2)$ space for these meteorological regimes (Fig. 8). Given the composition information available for the DI samples, consider the $\sigma_{DI-abs}$ panels in Figs. 7 and 8. The m/z 44:43 ratio suggests the DI soluble absorbers during the Stagnant period were aged OOA. This might suggest absorbing aerosol components capable of longer wavelength absorption were photochemically degraded to below detection values. The more limited photochemistry under the Transport/Haze period may have allowed such components to persist downwind long enough to be sampled at the *R/V Onnuri* position. Both the upwind and downwind Transport/Haze samples tended to exhibit low $\alpha_{DI-abs}$ values (Fig. 7) within a narrow range of $(a_1, a_2)$, but there were a few upwind samples with high $\alpha_{DI-abs}$ values on the partial spectra branch of the $(a_1, a_2)$ distribution (Figs. 7 and 8). All of the Stagnant and transition samples were on the partial spectra branch, while Blocking samples were split between the two branches. A detailed analysis of the organic composition of the DI extracts and the $(a_1, a_2)$ mapping of $\sigma_{DI-abs}$ is beyond the scope of this work,

but the evident separation in the maps shown here suggest future studies to relate composition to $\sigma_{DI\text{-}abs}$ may benefit from this approach.

### 3.4 Variability and spectral structure of $\omega(\lambda)$

While 2nd order polynomial coefficients provide more information than linear fits, additional spectral structure is evident in the complete spectra that cannot be captured by simple fits as in Sect. 3.3. In particular, $\omega$ spectra exhibited diverse shapes with a

range of features throughout the cruise (Fig. 10). The top 4 panels in Fig. 10 show $\omega(\lambda)$ calculated from the $\sigma_{ext}$ and $\sigma_{abs}$ spectra shown in Fig. 2. Two additional (bottom) panels are included to further illustrate the observed variability. The Transport/Haze - East 2 sample illustrates a relatively flat and featureless $\omega$ spectrum. The other two Transport/Haze examples show greater curvature in the UV portion of the spectrum. The Stagnant spectrum shows distinct UV features. The peak value of $\omega$ tends to be in the vicinity of 400 nm, but this is not always the case as evident in the Blocking 2 example (Fig. 10). The Blocking 2 example

is particularly intriguing for its mid-visible features, as well as its UV features.

It is important to note that the two broad UV features in the Stagnant example in Fig. 10 were present in all 4 Stagnant samples and were present to a lesser degree in the transition samples. They were absent in the Transport/Haze - East samples. Further, the features in the spectra of samples collected to the west of the peninsula (during both the Transport/Haze - West and Blocking

periods) differed from those to the east. Hence, while we noted in Section 2.3 that the measured absorbance spectra in the UV were noisy for wavelengths < 350 nm, the consistency evident in samples for each meteorological period indicate the features in Fig. 10 reflect ambient aerosols rather than an artifact of the measurement system. It is also particularly striking that the 2nd UV feature in the Stagnant samples exhibits a minimum at ~ 360 nm (outside of the noisy region of the deuterium lamp) where one would expect a signal from brown carbon consistent with the discussion in Section 3.2. All of the spectra were blank corrected

and only those above the lower limit of detection (LLOD = the mean blank spectrum + 3 standard deviations of the mean) were analyzed. It may be possible that spectral noise could manifest in a $\sigma_{abs}$ (and hence, $\omega$) spectrum if the values are near the LLOD, however, the appearance of similar features spanning a relatively broad range in wavelengths (compared to the noisy 0.2 nm resolution measurement for $\lambda$ < 350 nm) and appearing in both the UV and visible regions in multiple samples of a given meteorological period suggests the observed features were not attributable to noise. Future studies where composition

measurements that encompass the total aerosol and not just the water-soluble components are anticipated to enable the identification of specific molecular structures and/or compounds responsible for the kinds of features revealed in Fig. 10.

The variability and spectral features found in the $\sigma_{abs}$ (Fig. 2) and $\omega$ (Fig. 10) spectra suggest that spectral analysis tools such as peak fitting, as well as curvature coefficients (as in Sect. 3.3), may be useful in deriving new relationships between ambient aerosol

composition and optical properties. However, whatever mathematical approaches may prove useful, what Fig. 10 primarily reveals is the need for hyperspectral measurements to capture the variability present in $\sigma_{abs}$ and $\omega$ spectra of ambient aerosol populations. Neither power laws nor 2nd order polynomials can reproduce the features observed in this data set.

### 4 Conclusions

Parts 1 and 2 of this work have explored the information content of in situ hyperspectral measurements of ambient aerosols over

300-700 nm. Here in Part 2, the analyses focused on filter-based measurements of aerosol total absorption coefficients ($\sigma_{abs}$) and

the soluble absorption coefficients measured from DI and MeOH extracts ($\sigma_{DI\text{-}abs}$ and $\sigma_{MeOH\text{-}abs}$, respectively). $\sigma_{abs}$ together with the extinction coefficient ($\sigma_{ext}$) spectra (as reported in Part 1) averaged over the filter sampling intervals enabled calculation of hyperspectral single scattering albedo ($\omega$). Transforming the measured coefficient spectra into logarithmic space, it was found that all were better fit with 2$^{nd}$ order polynomials than with lines as would be expected from power law representations. The derivatives of the two fits are equivalent to each other such that any given Ångström exponent, $\alpha$, maps into a line in ($a_1$,$a_2$) coefficient space with a slope of $-2LN(\lambda_{ch})$, where $\lambda_{ch}$ is the characteristic wavelength of the measured spectrum at which the two fits yield the same result. This two-dimensional mapping space allows for separation among aerosol populations that otherwise exhibit the same $\alpha$. Schuster et al. (2006) explored the implications of this for bimodal aerosol size distributions with differing proportions of fine and coarse mode aerosols using ambient total column AERONET retrievals. Here, this method was applied to in situ measurements of the fine fraction only. Although size distributions were not measured with the in situ aerosol instrument suite aboard the *R/V Onnuri*, the published literature on the synoptic meteorology and its role in the observed regional characteristics of PM$_{2.5}$ (Peterson et al., 2019; Jordan et al., 2020a) permitted an assessment of the optical attributes reported here as a function of the varying particle sizes and composition encountered during the cruise. Clear separation of these ambient aerosol populations in ($a_1$,$a_2$) maps of hyperspectral optical properties indicate that such mapping offers more detailed information on the linkages between ambient aerosol optical properties and their chemical and microphysical characteristics than $\alpha$ alone can provide.

The May-June 2016 KORUS-OC cruise was a collaborative research campaign involving atmospheric composition and ocean color scientists within the broader umbrella of the KORUS-AQ mission. That collaboration across disciplines inspired the application of an established technique in the ocean color community to measure hyperspectral $\sigma_{abs}(\lambda)$ over the 300 - 700 nm range for in situ ambient aerosols. At the outset, there was some uncertainty as to the concentrations of aerosols likely to be encountered in the marine boundary layer throughout the cruise. That, along with limited space for personnel aboard ship, led to the decision to use relatively long filter sampling times (3 hr daytime, 12 hr nighttime) to ensure above detection samples were obtained on a reasonable sampling schedule. The separation in filter mean $\sigma_{ext}$ in ($a_1$,$a_2$) space for the differing aerosol populations was somewhat less distinct here than that shown in the higher temporal resolution spectra set of Part 1. This result suggests that improved temporal resolution in $\sigma_{abs}$ sampling may also lead to more distinct separation of different aerosols in ($a_1$,$a_2$).

Evident separation in the soluble absorption coefficients arose largely from the different parameterization of partial and complete spectra. The data were blank corrected and only those data above the lower limit of detection were reported and analyzed. All of the $\sigma_{abs}(\lambda)$ spectra were complete (i.e., spanned the full 300 - 700 nm wavelength range), but the soluble absorption spectra depended on the wavelength-dependent absorption of the soluble chromophores, such that for some samples only partial spectra of the soluble extracts ($\sigma_{DI\text{-}abs}(\lambda)$ and $\sigma_{MeOH\text{-}abs}(\lambda)$) were above the detection limit. Partial spectra led to larger values in $\alpha$ and more extreme values in ($a_1$,$a_2$), both positive and negative.

The DI extracts were also analyzed for water-soluble aerosol composition allowing for direct comparison of water-soluble composition to the optical properties. Examples of relationships for the optical properties at 365 nm were examined and found to differ such that $\sigma_{ext}(365$ nm) was best correlated with anthropogenic pollution tracers (SO$_4^{2-}$ and NH$_4^+$), both $\sigma_{DI\text{-}abs}$ and $\sigma_{MeOH\text{-}abs}$ were best correlated with oxalate (C$_2$O$_4^{2-}$), while $\sigma_{abs}(365$ nm) was not well correlated with any water-soluble ion. These are not unexpected results. More interestingly, elevated C$_2$O$_4^{2-}$ could be separated into 2 groups, one accompanied by elevated K$^+$ and one that was not. Both groups were observed downwind of the Korean peninsula, with the former group present during the

Transport/Haze period and the latter present during the Stagnant period. m/z 43, 44, and 60 from AMS measurements of the nebulized DI extracts were examined to shed more light on these populations. m/z 60 revealed no evidence of biomass burning in the data set as might be expected from elevated $K^+$ and $C_2O_4^{2-}$. Higher ratios of m/z 44:43 observed for the Stagnant group suggested this group was more photochemically aged compared to the Transport/Haze group, consistent with what is known of the synoptic meteorology of those periods (Peterson et al., 2019). The soluble absorption spectra for the Stagnant group were partial

which may indicate chromophores that might have absorbed at longer wavelengths had degraded below detection, another potential indication of photochemical aging. Further examination of these two groups of samples using $\omega(365\ nm)$ and $\omega(532\ nm)$ showed the Stagnant group exhibited a much larger decrease in $\omega$ from 532 nm to 365 nm than the Transport/Haze group. This result suggests that elevated $C_2O_4^{2-}$ can represent different chromophores within the soluble organic aerosol population. Hence, it ought not to be inferred that the two groups differed only due to aging, the underlying chromophores may have arisen from different

sources and/or atmospheric processes entirely, but photochemical aging may have been one factor contributing to the differences.

Finally, it was shown that spectral fit parameters (whether from power laws or 2$^{nd}$ order polynomials) cannot capture spectral features that were observed in $\sigma_{abs}$ and $\omega$. These features highlight the need for hyperspectral measurements to capture the variability of optical properties in ambient aerosols. Spectral analysis tools such as peak fitting applied to measured spectra may

be useful in further exploration of the wavelength-dependence of chromophores.

From the perspective of in situ ambient sampling alone, it is anticipated that hyperspectral $\sigma_{ext}$, $\sigma_{abs}$, $\sigma_{DI-abs}$, and $\sigma_{MeOH-abs}$ coupled to commensurate composition and size distribution information will lead to advances in deriving the physicochemical properties of ambient particles from optical data sets. Further, it is also expected that such measurements will be needed to complement

planned hyperspectral remote sensors for upcoming satellite missions such as Tropospheric Emissions: Monitoring of Pollution (TEMPO, Zoogman et al., 2017) and Phytoplankton, Aerosol, Cloud, Ocean Ecosystem (PACE, Werdell et al., 2019). TEMPO splits its hyperspectral range into two parts: 290-490 nm and 540-740 nm each with a spectral resolution of 0.57 nm (Zoogman et al., 2017). The Ocean Color Imager (OCI) planned for PACE will cover 340-890 nm at 5 nm resolution (plus 7 additional bands extending further into the IR range) and it will be joined by the SPEXone polarimeter spanning 385 - 770 nm at 2-4 nm resolution

for that satellite (Werdell et al., 2019). It will be necessary to have in situ measurement capabilities for validation of such remote sensors. The in situ spectra reported in this work illustrate how hyperspectral information can provide new approaches that may be explored to expand the suite of aerosol products that can be retrieved from hyperspectral remote sensors.

**Data Availability**

All data presented here are available under the *R/V Onnuri* Ship tab in the KORUS-AQ archive (DOI:
10.5067/Suborbital/KORUSAQ/DATA01).

**Author contribution**

CEJ led the experiment, analyzed the data, and wrote the manuscript.
CEJ, BEA, LDZ, CHH, KLT, ELW, RFM, MAS, AJB, CER built elements of the hardware, software, and deployment measurement system; and assisted in the laboratory at NASA LaRC.
CEJ, BEA, AJB, CAC participated in the field work.

RMS, BTL, & MAT deployed with the measurement suite aboard the *R/V Onnuri*, collected filter samples, and contributed to the manuscript.

MN, AM, BEA, ECC, ELW, LDZ, & CEJ, provided laboratory equipment and contributed to post-cruise filter analyses at NASA GSFC and LaRC.

AM, GLS, RHM, LDZ, BEA, ECC, MAS, AJB, & CAC contributed to the data analysis and the manuscript.

**Competing interests**

The authors declare that they have no conflict of interest.

**Acknowledgments**

The authors gratefully acknowledge the support of the KORUS-OC and KORUS-AQ science teams, the outstanding support
provided by our South Korean partners at the Korean Institute for Ocean Science and Technology (KIOST), and financial support from the NASA/NIA cooperative agreement NNL09AA00A and NASA Grant NNX16AD60G through the Geostationary Coastal and Air Pollution Events (GEO-CAPE) mission pre-formulation studies. The authors particularly thank Anne Thompson for her support throughout this study and Aimee Neeley and Ryan Vandermeulen for helpful discussions.

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

**Table 1.** Summary of measurements in this study with estimated measurement errors.

| Measurement Method | Parameter | Estimated Measurement Error |
|---|---|---|
| in situ Spectral Aerosol Extinction Instrument (SpEx) | Aerosol extinction coefficients ($\sigma_{ext}$) | ~ 5% (300-700 nm)[a] |
| AirPhoton IN101 nephelometer | Aerosol scattering coefficients ($\sigma_{scat}$) at 450, 532, and 632 nm | < 1 Mm$^{-1}$ (all wavelengths)[b] |
| Brechtel Tricolor Absorption Photometer (TAP) | Aerosol absorption coefficients ($\sigma_{abs}$) at 467, 528, and 652 nm | ~ 0.2 Mm$^{-1}$ (based on 60 s means, all wavelengths)[c] |
| Glass fiber filter in integrating sphere attached to dual beam spectrophotometer | Total aerosol absorption coefficients ($\sigma_{abs}$) | ~ 17% (300-700 nm)[d] |
| Teflon filter extracts in liquid waveguide capillary cell using two solvents: | | |
| methanol (MeOH) | MeOH-soluble aerosol absorption coefficients ($\sigma_{MeOH-abs}$) | ±30%[e] |
| deionized 18 MΩ water (DI) | DI-soluble aerosol absorption coefficients ($\sigma_{DI-abs}$) | ±30%[e] |
| Ion chromatography of DI extracts | $NH_4^+$, $K^+$, $Ca^{2+}$, $Mg^{2+}$, $Na^+$, $SO_4^{2-}$, $NO_3^-$, $Cl^-$, and $C_2O_4^{2-}$ (mass & equivalents concentrations) | ±30%[e] |
| Aerosol mass spectrometry of DI extracts | Organics, sulfate, nitrate, ammonium, and chloride (relative contributions to a sample) | ±30%[e,f] |

[a]*Jordan et al., 2015; 2020b.*
[b]*Operating Manual for Sampling Station and Nephelometer, Version 1.1, Feb. 2014 (AirPhoton, Baltimore, MD).*
[c]*Brechtel TAP Tricolor Absorption Photometer Model 2901 TAP System Manual Revision 1, June 24, 2015 (Brechtel, Hayward, CA)*
[d]*estimated from propagation of uncertainty as discussed in Section 2.3.*
[e]*uncertainty in the extraction volume was the dominant term in error propagation leading to somewhat greater than typical estimated errors in the data from solvent extracts*
[f]*as discussed in Section 2.5 it is not possible to derive mass concentrations for the AMS data reported here, hence, the estimated error reflects that of the other solvent extract data for consistency*

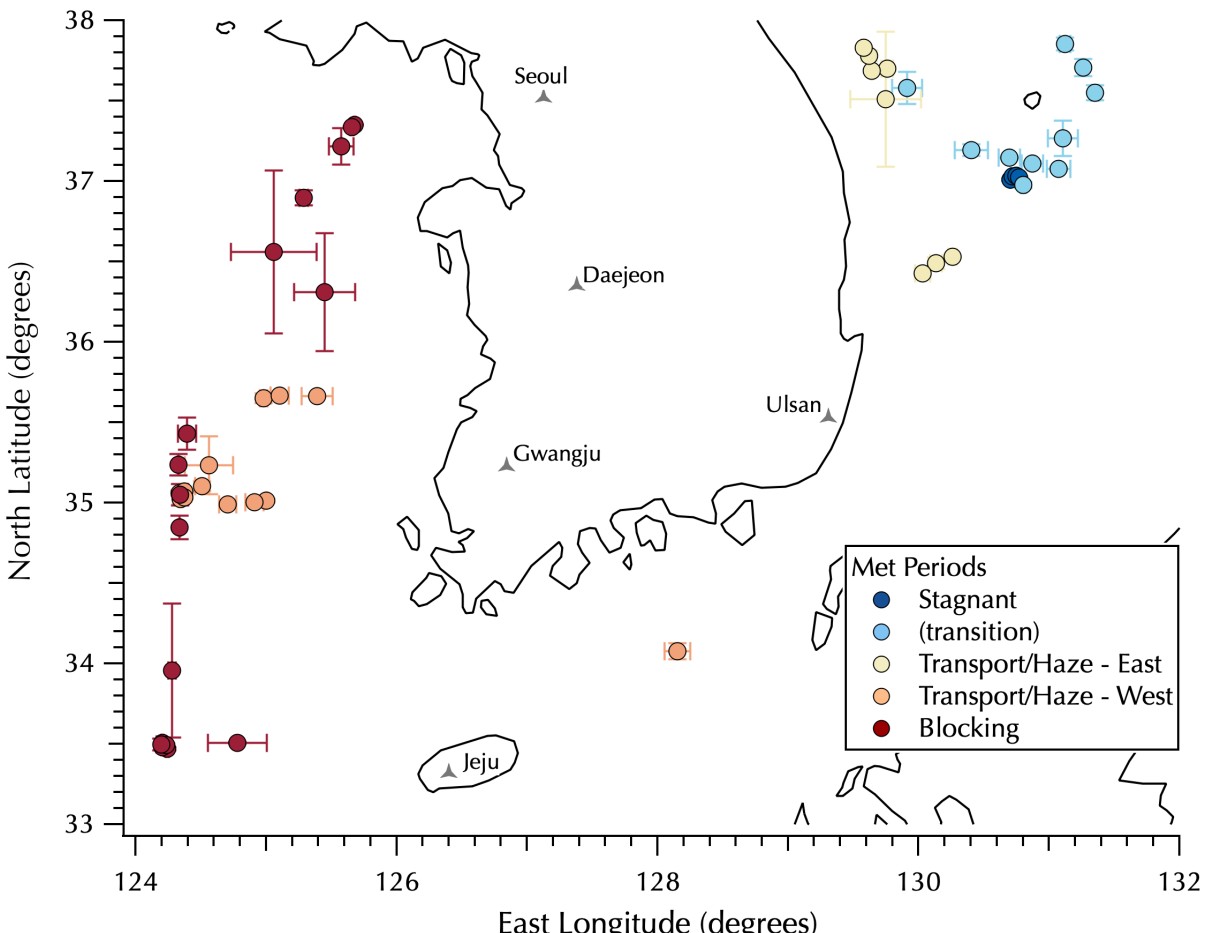

**Figure 1: Map of mean sampling locations for the filter data set colored by the Peterson et al. (2019) meteorological regimes with Transport/Haze data split between upwind (west, orange) and downwind (east, yellow) positions around the peninsula. Error bars represent one standard deviation of the mean filter sample latitude and longitude, used to distinguish filters collected on station from those collected underway. See Table S1 for details.**

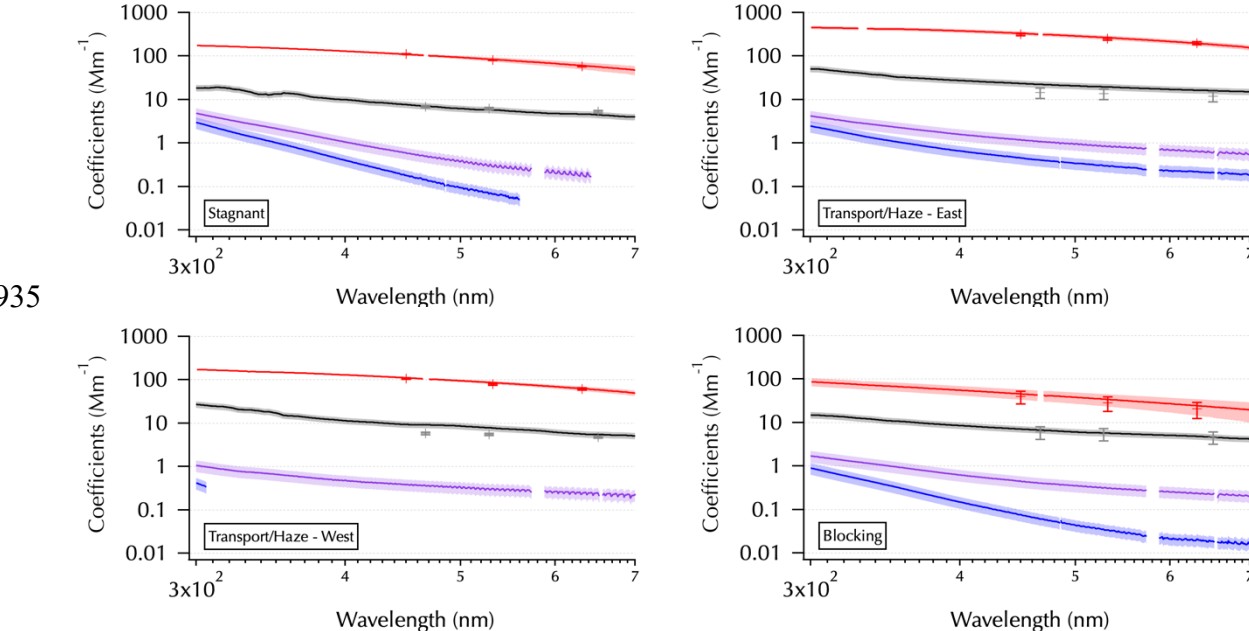


**Figure 2: One example from each meteorological regime as shown in Fig. 1 of the spectra set obtained from each filter: mean SpEx σ_ext (red line, shaded with ± 1 standard deviation), σ_abs (black line), σ_MeOH-abs and σ_DI-abs (purple and blue lines, respectively). Measurement error for all of the absorption coefficient spectra estimated using error propagation (shading, on log scale this is more difficult to discern**

**for σ_abs than for σ_MeOH-abs and σ_DI-abs). Mean NT σ_ext (red symbols ± 1 standard deviation) and mean TAP σ_abs (gray symbols ± 1 standard deviation) are shown for comparison. All in units of Mm⁻¹.**

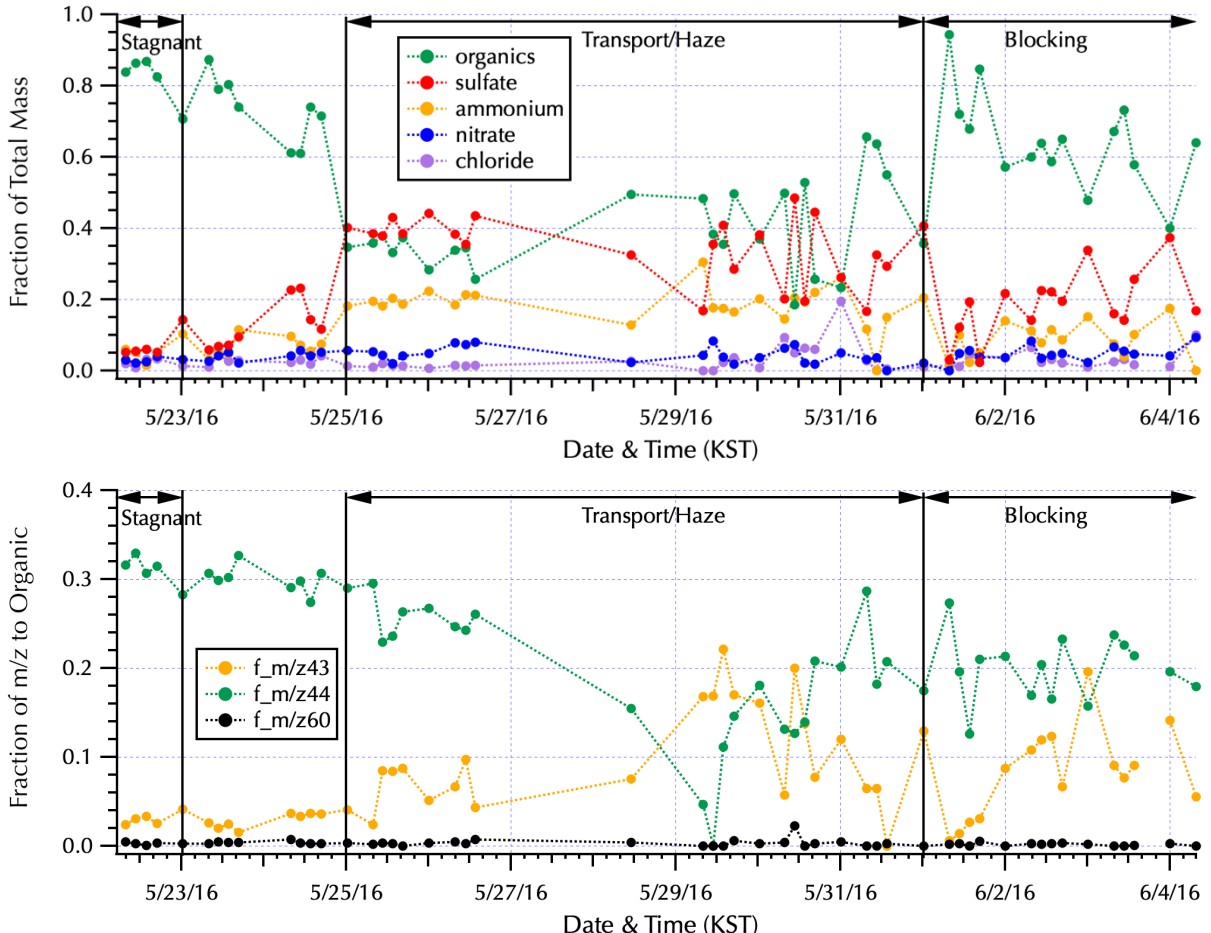

**Figure 3:  The fraction contributed to the total mass measured by AMS by each of the major aerosol components (top panel) and the fraction of key water soluble organic mass fractions to the total AMS organic mass (bottom panel) for the complete filter set (Table S1) are shown within the context of the Peterson et al. (2019) meteorological regimes.  Note, the long gaps between filters during the Transport/Haze period occurred as the ship transited South Korea's territorial seas with samples obtained when the ship was outside of those waters.  The single filter obtained along the south coast (Fig. 1) is the isolated data point on May 28[th].  The Transport/Haze samples**
**collected from May 25[th]-26[th] were obtained to the east of the peninsula, those from May 29[th]-31[st] were obtained to the west, representing downwind and upwind samples with respect to South Korean emissions and local atmospheric processing, respectively, under the prevalent atmospheric transport of this meteorological period..**

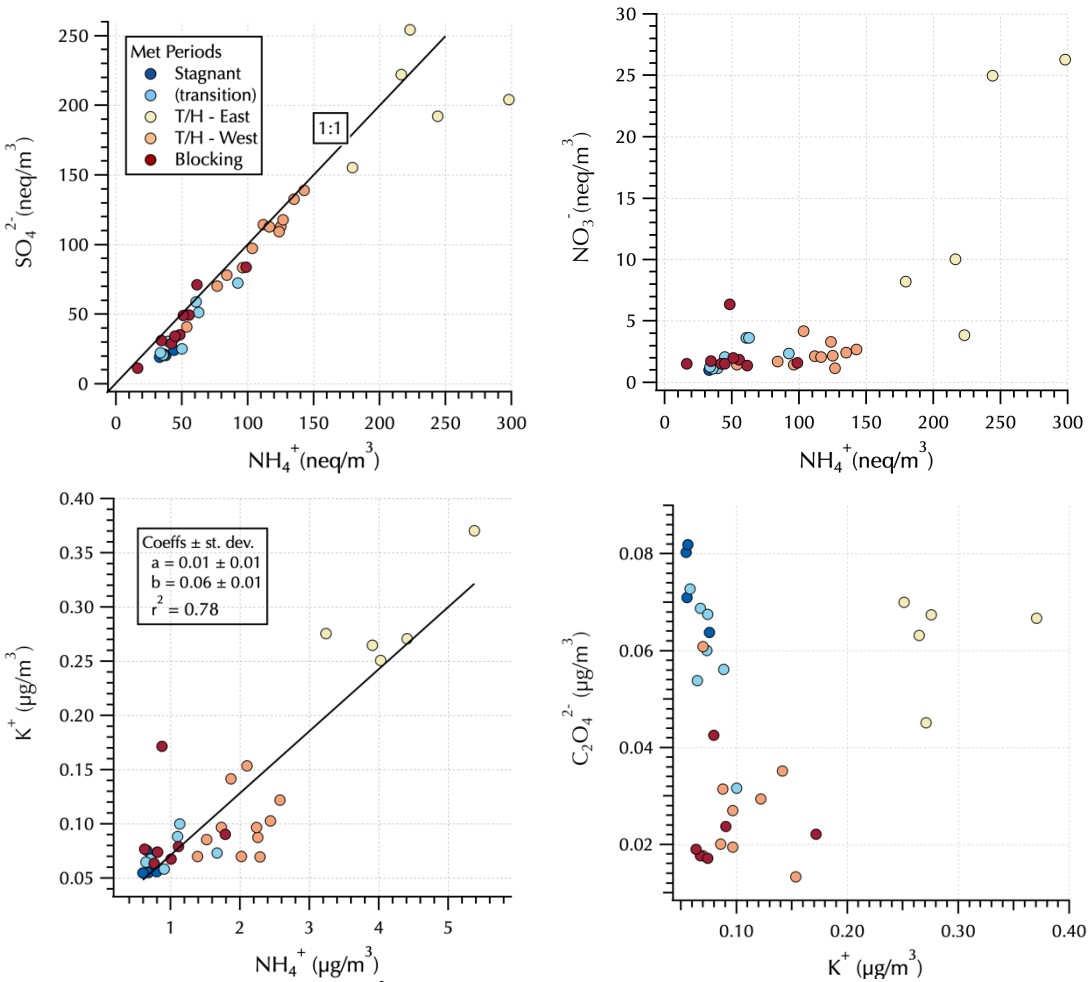


**Figure 4:** Relationships between $SO_4^{2-}$ and $NH_4^+$ (top left) and $NO_3^-$ and $NH_4^+$ (top right) in equivalents units, and between $K^+$ and $NH_4^+$ (bottom left) and $C_2O_4^{2-}$ and $K^+$ (bottom right) in mass units. Markers are colored by Peterson et al. (2019) meteorological regime with the Transport/Haze - East samples downwind of South Korea and the Transport/Haze - West samples upwind of South Korea (see Fig. 1) under the prevailing low level westerly transport from China during this meteorological regime.


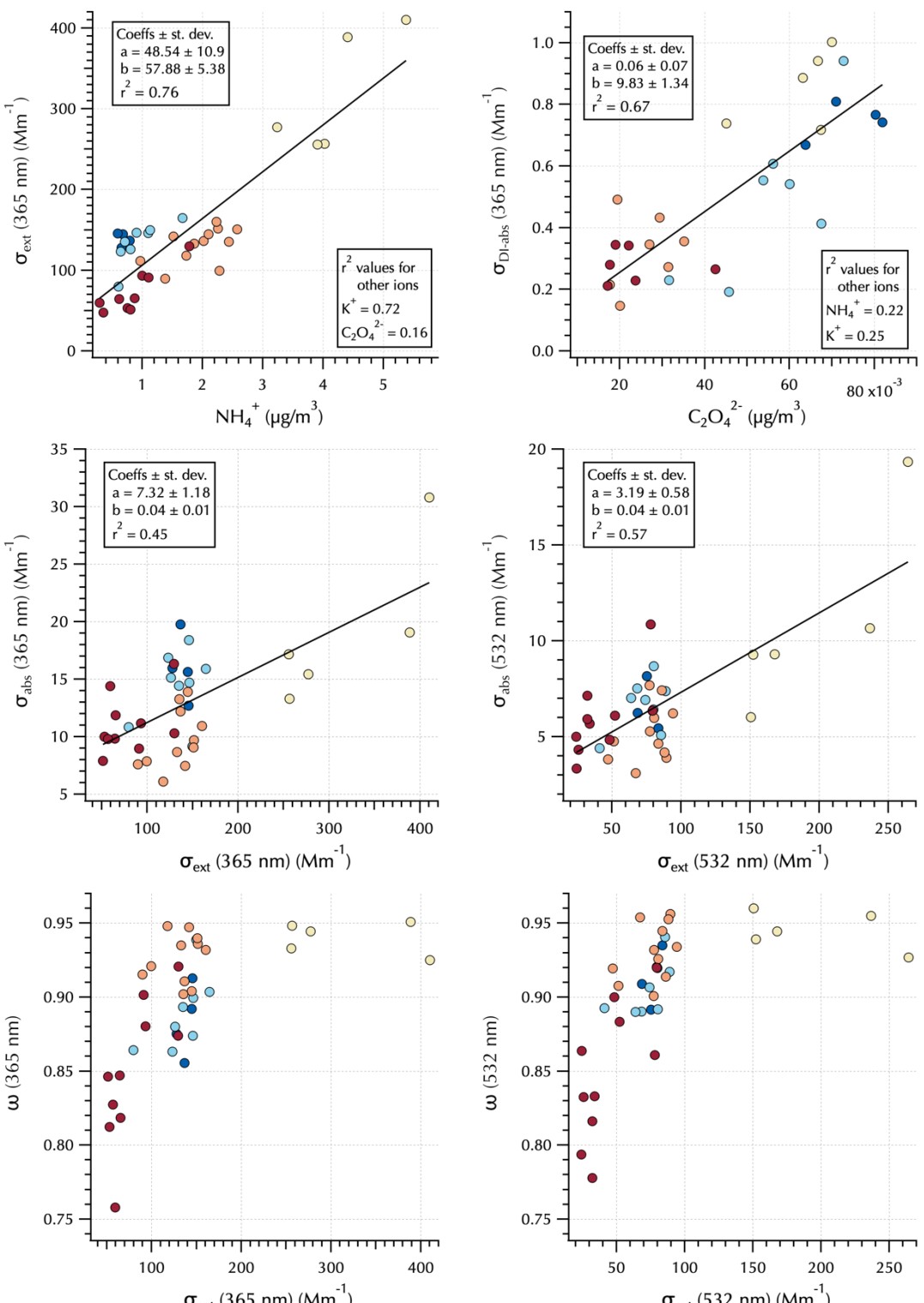

**Figure 5: Relationships between 365 nm $\sigma_{ext}$ and $NH_4^+$ (top left), 365 nm $\sigma_{DI-abs}$ and $C_2O_4^{2-}$ (top right), $\sigma_{abs}$ and $\sigma_{ext}$ at 365 and 532 nm (middle left and right, respectively), and $\omega$ and $\sigma_{ext}$ at 365 and 532 nm (bottom left and right, respectively). Markers are colored by Peterson et al. (2019) meteorological regime (see Figs. 1 and 4).**

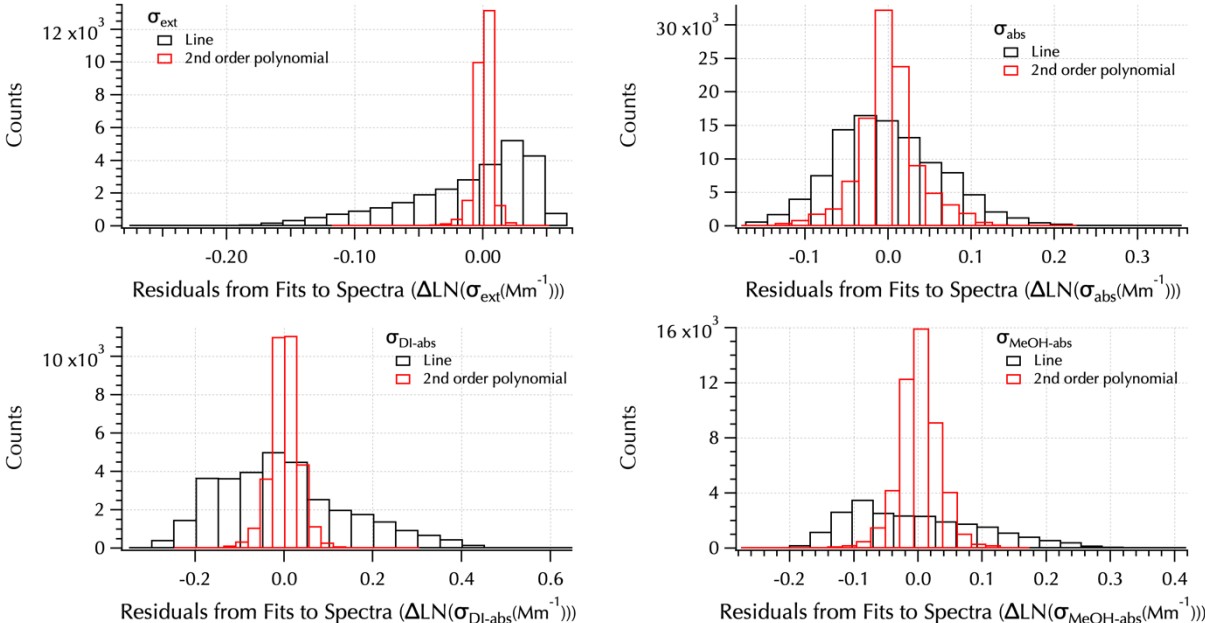

**Figure 6: Residuals from linear (black) and 2$^{nd}$ order polynomial (red) fits to each spectra set: $\sigma_{ext}$ (top left), $\sigma_{abs}$ (top right), $\sigma_{DI\text{-}abs}$ (bottom left), $\sigma_{MeOH\text{-}abs}$ (bottom right).**

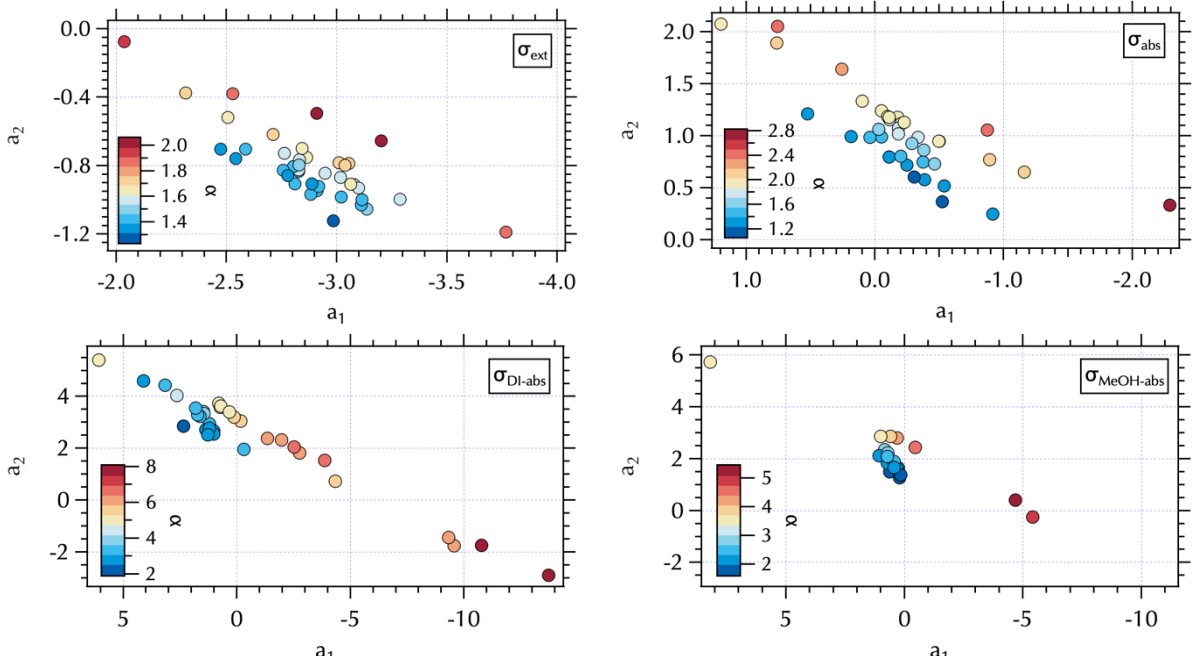

Figure 7: Maps of α in a₂ vs. a₁ space: σ_ext (top left), σ_abs (top right), σ_DI-abs (bottom left), and σ_MeOH-abs (bottom right).


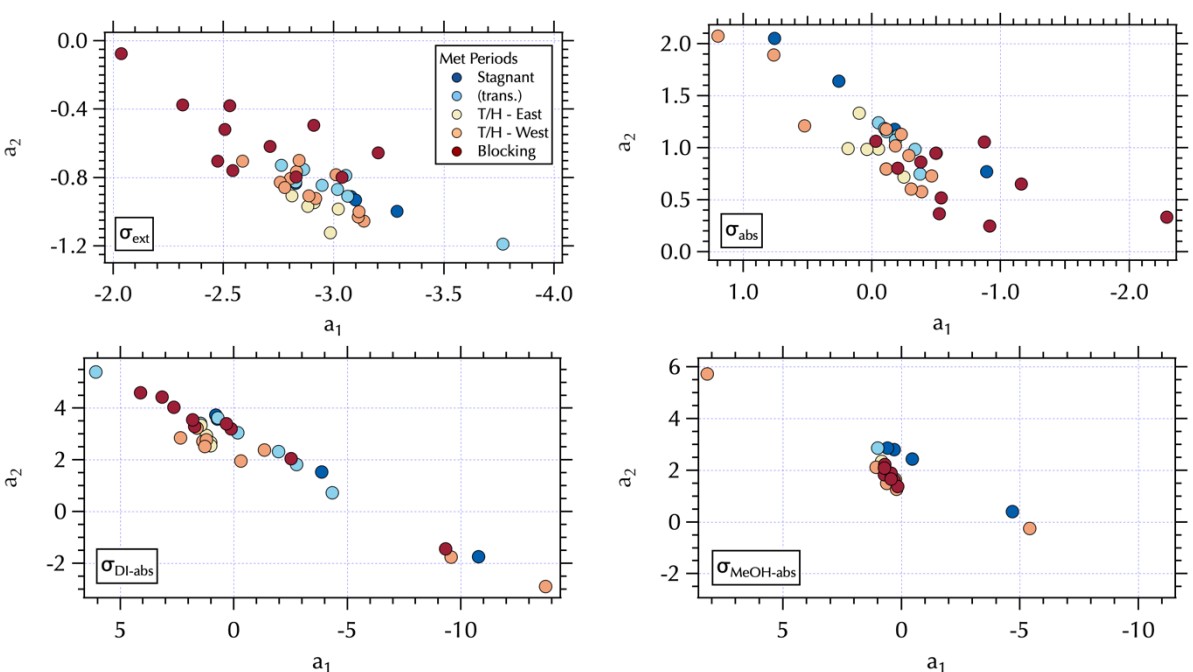

Figure 8: Maps of the meteorological groups as in Fig. 1 in a₂ vs. a₁ space: σ_ext (top left), σ_abs (top right), σ_DI-abs (bottom left), and σ_MeOH-abs (bottom right).


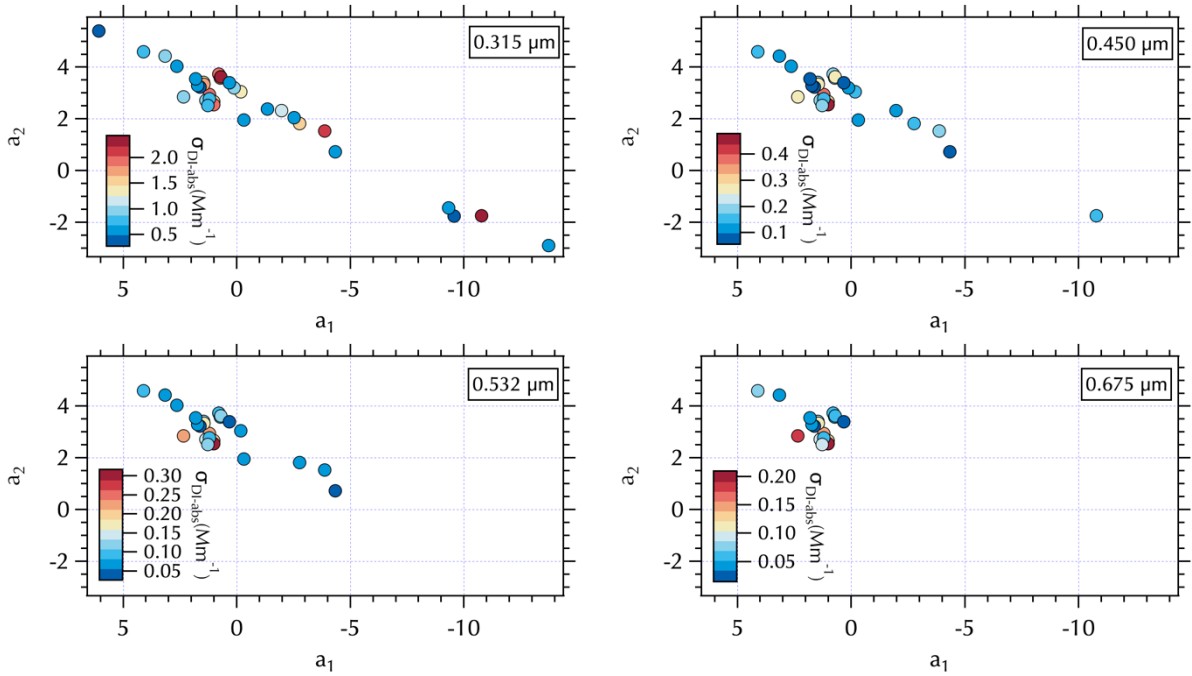

**Figure 9: Maps of $\sigma_{DI\text{-}abs}$ (Mm$^{-1}$) in $a_2$ vs. $a_1$ space at: 0.315 µm (top left), 0.450 µm (top right), 0.532 µm (bottom left), and 0.675 µm (bottom right).**


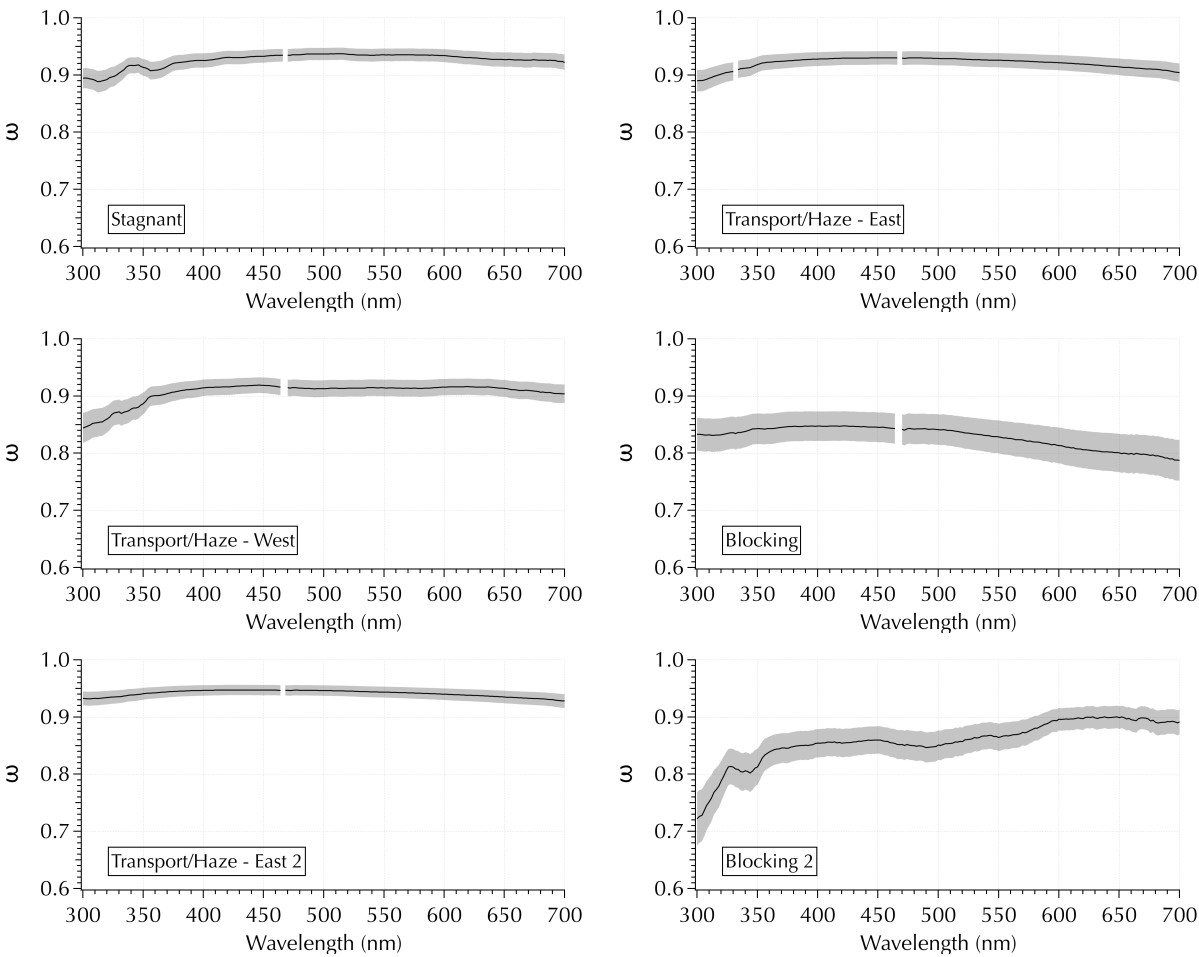

Figure 10: Examples of variability in ω spectra using the same four examples as in Fig. 2 (top four panels), plus an additional example from downwind Transport/Haze and Blocking (bottom two panels, respectively).