# Peer review of "New In Situ Aerosol Hyperspectral Optical Measurements over 300-700 nm, Part 2: Extinction, Total Absorption, Water- and Methanolsoluble Absorption observed during the KORUS-OC cruise"

_Atmospheric Measurement Techniques, 2020_

## Referee Comment (RC1) · Anonymous Referee #1 · 17 Sep 2020

General comments:

This manuscript presents valuable spectral absorption (and extinction) spectrum of aerosols covering ultraviolet (from 300 nm) measured over the ocean nearby Korean peninsula. This manuscript is generally well-written and clearly described their measurements. Hence, I recommend publication of this manuscript after considering few of my suggestions, which I believe can clarify statements and attract a broader com-

munity. As the authors stated, the spectral extinction and absorption cross-sections are affected by both particle size distribution and their physicochemical properties (shape and complex refractive indices). However, in my understanding, the spectral extinction is more weighted by particle size, whereas spectral absorption features are more affected by their chemical characteristics, which makes notable differences from Part-1 paper they submitted. In particular, unlike the extinction cross-section, spectral absorption by aerosols in the UV are known to have distinct features, which are supposed to be hard to be extrapolated from longer wavelengths. In addition, absorption by aerosols in the UV has particular importance in many reasons (e.g., Zhang et al., 2013, 2017 in their reference list and Mok et al., 2016 and references therein; https://doi.org/10.1038/srep36940), but yet suffers from lack of reliable measurements over globe. The spectral variability in the UV does appeared in their measurements (Figure 10). Therefore, I believe elaborating UV aerosol absorption, together with more discussions of UV-specific error estimations of the spectral features (e.g., out-of-band stray light of the spectrometer, and stability of light source in the UV), likely at Section 3.4, can even more emphasize the values of this study.

Specific Comments:

Abstract at lines 36-37: As I suggested in the general comments, I think it worth to note that the measurement captured detailed spectral features of the single scattering albedo of aerosols, other than describing limitations of 2nd-order polynomial fitting. But it is up to the authors weather reflect this comment or not.

This study utilizes several instruments including ion chromatography and aerosol mass spectrometry in addition to their main instruments. I think a brief table in the main script summarizing these instruments (measurement method, target, their estimated error) can help readers to easily understand the measurements.

Lines 175-177: Why the UV portion was noisy? Is it due to the relatively low level of intensity of the light source in the UV or stray light? Is there any possibility that noise

could propagated to the spectral features in the UV in Figure 10?

Line 213: Please elaborate the "error propagation".

Lines 315-316: Do you have any explanations of 'smoothly varying" and "spectral feature" events?

Technical Corrections:

Reference Section: The part-1 paper (Jordan et al., 2020b) supposed to be submitted to AMT.

––––––––––––––––––––––––––––––––

---

## Referee Comment (RC2) · Anonymous Referee #2 · 13 Oct 2020

**Review on "New In Situ Aerosol Hyperspectral Optical Measurements over 300- 700 nm, Part 2: Extinction, Total Absorption, Water- and Methanol- soluble Absorption observed during the KORUS-OC cruise" by Jordan et al.**

**General Comments:**

The manuscript provides and discusses the retrievals of aerosol absorption and SSA over a wide wavelength range of 300-700 nm obtained from in situ sampling methods employed during the May-June 2016 Korea United States - Ocean Color (KORUS-OC) cruise conducted in concert with the broader air quality campaign (KORUS-AQ). Various filter-based measurement techniques are applied to obtain total aerosol absorption and soluble-absorption coefficients. Authors further used the extinction retrievals from the Part-I of this study and combined it with the absorption retrievals calculated here to obtain the SSA values. The measurements/retrievals were further put in context of the expected aerosol composition for the different meteorological regimes observed over the period of the campaign. The findings of this study are valuable to the aerosol community because absorption retrievals in the near-UV range are sparse in general and recent studies have highlighted the importance of quantifying the aerosol absorption in the near-UV to visible wavelengths, wherein presence of brown carbon/organic carbon show a steep spectral dependency. However, this study particularly lacks in detail explaining of the absorption features in this important near-UV wavelength range, either due to the noise in the measurement spectra itself or due to the way correction factors for higher wavelength range are extrapolated for these smaller wavelengths. This issue can be addressed by providing clarification statements wherever required, added speculations for observed features in near-UV range and better quantifying the errors associated with the total absorption retrievals in the near-UV wavelength range. I recommend the publication of this manuscript after following suggestions/comments are incorporated.

**Specific/major comments:**

Section 2.1: It is okay to reference Peterson et al. 2019 here for the classification of meteorological regimes that are used in the subsequent analysis, however, a brief description of what each regime means within this section would be more reader-friendly (instead of waiting for this clarification in section 3.2. later). For example, 'Blocking' term/regime is not self-explanatory. This hinders the understanding of Fig. 1 as well (for a reader) that is associated with the text in this section.

1. Line 146: "These data were retained for the assessment of the correction factor for the spectral absorption measurement discussed in Sect. 2.3". Can the authors discuss the rationale behind this choice? What implications or bias do they anticipate of including the contaminated measurements in their subsequent assessment of correction factor?

2. Section 2.3: How accurate or reliable is the use of the correction factor, $\beta$, based on $\sigma_{abs}$/Abs at three specific wavelengths (467, 528, and 652) of TAP in calculating the $\sigma_{abs}$ outside of this wavelength range, specially at UV and near-UV wavelength ranges that are much smaller than 467 nm? Also, suggest moving Figure S3 to being part of the main text, and adding the final $\beta_{K-OC}$ that was obtained after scaling.

3. Based on the above point 4, what impact will the uncertainty in estimation of $\beta$ for near-UV wavelengths, combined with the noisy absorbance measurements in the same wavelength range, have on the overall quality of retrievals for $\sigma_{abs}$ using this filter-retrieval method? As understood by the aerosol community and the authors also point out elsewhere in the manuscript, retrievals in this wavelength-range are particularly important for the estimation of absorption by brown carbon/organic carbon aerosol particles. The manuscript needs more elaborate discussions on this aspect.

4. Line 213-14: This sentence on error propagation is hard to understand from the way it is currently formulated. From equation 1, apart from the uncertainty in absorbance, there is also an uncertainty associated with scaling of $\beta$ and may be further in the extrapolation of $\beta$ for smaller wavelengths (if at all accounted for).

5. Section 3.1: I wonder if this section should be discussed in the light of discussions that follow regarding the differences in aerosol composition and size between the different meteorological regimes for better understanding of the differences between the absorption spectra obtained from the filters using different methods. For example, it is hard for the readers to move forward with the reading while questions about, why $\sigma_{MeOH-abs}$ is above detection, while little of the $\sigma_{DI-abs}$ is above detection for the Transport/Haze example lingers on. It would be nice to either include the speculated reasons for these differences within this section and explain the reasons for these speculations further in the following sections or rather discuss all of the absorption spectra differences in the light of what we know about the composition and size distribution of aerosol particles in each meteorological regime. Some of the differences in Figure2, mentioned in section 3.1 are not even addressed later on.

6. Section 3.4: Figure 10 and this section is possibly one of the most useful sections for aerosol community as measurements/retrievals of SSA ($\omega$) across a wide range of wavelengths (300-700 nm) can provide a significant constraint on aerosol modeling efforts. Suggest highlighting this part in the abstract for larger appeal. Also, spectral variation in SSA in near-UV range for multiple instances/examples in Figure 10 is interesting. How much of this variation is attributed to the uncertainty in retrieval methods for this particular wavelength range and how much of this is a real feature? This should be discussed in greater detail, rather than leaving this part just as an observation and proposing some curve-fitting exercise.

**Minor/Editorial comments:**

Line 109: Is 'package' the best suitable word to describe a suite of instruments? 'Instrument suite' or something similar sounds more appropriate.

Line 27: It is not specified previous to this sentence what DI stands for.

Line 46: "......aerosol size distributions than cannot be obtained from single Ångström exponents alone.

Line 51: than Ångström exponents (AE)-> compared to the AE approach

Line 54: This sentence is confusing. Suggested change: The data reported in this study were from in situ aerosol measurements in a package deployed aboard the ...-> The data reported in this study are in situ aerosol measurements from a suite of instruments deployed aboard the....

Line111: ..... influenced the observed spectral characteristics observed aboard the R/V Onnuri. -> influenced the observed spectral characteristics obtained from the instruments aboard the R/V Onnuri.

Line 121: Insert a Figure 1 reference right here.

Line 122: ....50% size cut of 1.3 µm diameter. -> 50% cut-off size of 1.3 µm diameter.

Line 139: "The high-resolution data set was 'filtered' to remove interceptions of the R/V Onnuri's own ship stack emissions (Jordan et al., 2020b)". I think the use of term 'filtered' here can be confusing. Suggest using 'screened' or another synonym.

Line 164: OC -> Ocean Color (first time occurrence of OC as a noun)

Line 346: Is there a way to identify the downwind and upwind side of the peninsula in Figure 1 (e.g. plotting MERRA-2 (or any other reanalysis) wind vectors over the map). It is hard to understand which part of the plot is being refereed as downwind or upwind in Figure 3 and Figure 4 as well.

---

## Author Response (AR1)

***Response to referees' comments on*** "New In Situ Aerosol Hyperspectral Optical Measurements over 300–700 nm, Part 2: Extinction, Total Absorption, Water- and Methanol-soluble Absorption observed during the KORUS-OC cruise" ***by*** Carolyn E. Jordan et al.

On behalf of my co-authors and myself, I would like to thank the referees for their thoughtful reviews of this manuscript. Your insights have strengthened this work and we greatly appreciate your time and effort on our behalf. We have made most of the recommended changes as described in detail below and as is evident in the tracked changes revision of our submission. For consistency with the numbers used by the referees to call out places in the text in need of modification, we refer to those numbers in our responses with the updated numbers for the revised text provided in parentheses. Below, we show the original comments in gray italics, our replies in black regular font, and updated text copied over from the revision in black italics.

**Anonymous Referee #1**

*General comments:*

*This manuscript presents valuable spectral absorption (and extinction) spectrum of aerosols covering ultraviolet (from 300 nm) measured over the ocean nearby Korean peninsula. This manuscript is generally well-written and clearly described their measurements. Hence, I recommend publication of this manuscript after considering few of my suggestions, which I believe can clarify statements and attract a broader community. As the authors stated, the spectral extinction and absorption cross-sections are affected by both particle size distribution and their physicochemical properties (shape and complex refractive indices). However, in my understanding, the spectral extinction is more weighted by particle size, whereas spectral absorption features are more affected by their chemical characteristics, which makes notable differences from Part-1 paper they submitted. In particular, unlike the extinction cross-section, spectral absorption by aerosols in the UV are known to have distinct features, which are supposed to be hard to be extrapolated from longer wavelengths. In addition, absorption by aerosols in the UV has particular importance in many reasons (e.g., Zhang et al., 2013, 2017 in their reference list and Mok et al., 2016 and references therein; https://doi.org/10.1038/srep36940), but yet suffers from lack of reliable measurements over globe. The spectral variability in the UV does appeared in their measurements (Figure 10). Therefore, I believe elaborating UV aerosol absorption, together with more discussions of UV-specific error estimations of the spectral features (e.g., out-of-band stray light of the spectrometer, and stability of light source in the UV), likely at Section 3.4, can even more emphasize the values of this study.*

Thank you. Both you and Referee #2 had the insight for us to do more to emphasize the point of Section 3.4, which is that it is necessary to measure hyperspectral single scattering albedo (via extinction and absorption) in order to capture the spectral features evident in this data set, given that such features are likely to be found in ambient aerosol populations more broadly. You both also expressed concern about the UV noise in the measured absorbance that we have now

addressed in Section 3.4.  We have amended the text in the Abstract, Section 3.4, and the Conclusions to do a better job with this as you will see detailed below.  Thank you (both) again.

*Specific Comments:*

*Abstract at lines 36-37: As I suggested in the general comments, I think it worth to note that the measurement captured detailed spectral features of the single scattering albedo of aerosols, other than describing limitations of 2nd-order polynomial fitting. But it is up to the authors weather reflect this comment or not.*

Thank you for this excellent suggestion, we have revised the abstract replacing the sentence that started with "However" on line 36 as follows"

*"However, a key finding of this work is that mathematical functions (whether power laws or 2$^{nd}$ order polynomials) extrapolated from a few wavelengths or a subrange of wavelengths fail to reproduce the measured spectra over the full 300-700 nm wavelength range.  Further, the $\sigma_{abs}$ and $\omega$ spectra exhibited distinctive spectral features across the UV and visible wavelength range that simple functions and extrapolations cannot reproduce.  These results show that in situ hyperspectral measurements provide valuable new data that can be probed for additional information relating in situ aerosol optical properties to the underlying physicochemical properties of ambient aerosols."*

Further, we modified the relevant paragraph (starting at line 575 (662))in the Conclusions as follows:

*"Finally, it was shown that spectral fit parameters (whether from power laws or 2$^{nd}$ order polynomials) cannot capture spectral features that were observed in $\sigma_{abs}$ and $\omega$.  These features highlight the need for hyperspectral measurements to capture the variability of optical properties in ambient aerosols.  Spectral analysis tools such as peak fitting applied to measured spectra may be useful in further exploration of the wavelength-dependence of chromophores."*

*This study utilizes several instruments including ion chromatography and aerosol mass spectrometry in addition to their main instruments. I think a brief table in the main script summarizing these instruments (measurement method, target, their estimated error) can help readers to easily understand the measurements.*

Table 1 has been created and is cited in the Introduction, Section 2.3, and Section 3.1.

*Lines 175-177: Why the UV portion was noisy? Is it due to the relatively low level of intensity of the light source in the UV or stray light? Is there any possibility that noise could propagated to the spectral features in the UV in Figure 10?*

The UV noise is typical for this type of dual lamp spectrophotometer.  The instrument scans from the longest to shortest wavelengths using a stable (less noisy) tungsten (halogen) lamp for most of the spectral range, but the intensity drops off too much to use that light source for

wavelengths less than 350 nm.  At 350 nm, the spectrophotometer switches to a deuterium lamp, but it is noisy.  We have added text starting at line 175 (203) to explain this:

*"The tungsten (halogen) lamp provides a stable signal that spans most of the spectral range. Scanning from longer to shorter wavelengths at 350 nm the instrument switches to a much noisier deuterium lamp to cover the remainder of the UV range where the intensity of the tungsten light source is too low to make a good measurement.  To minimize the noise present in the UV, a 10 nm boxcar smoothing algorithm was applied to the full wavelength range of the 0.2 nm scans."*

Another contributing factor to the noise is scattering within the filter below 350 nm.  The ocean color community is working towards measuring amplification factors (i.e., the β correction) for particulates filtered from aquatic samples using a PSICAM that has recently been commercialized, but the lower UV will still be out of reach.  At present, amplification correction factors do not have a wavelength dependence, but perhaps that will change one day with further study targeting the UV range.  To address the final question here, we have added the following paragraph to the discussion of Fig. 10 in Section 3.4 starting at line 583 of the revised manuscript (rather than in the Methods section, lines 175-177 in Section 2.3 of the submitted manuscript):

*"It is important to note that the two broad UV features in the Stagnant example in Fig. 10 were present in all 4 Stagnant samples and were present to a lesser degree in the transition samples. They were absent in the Transport/Haze - East samples.  Further, the features in the spectra of samples collected to the west of the peninsula (during both the Transport/Haze - West and Blocking periods) differed from those to the east.  Hence, while we noted in Section 2.3 that the measured absorbance spectra in the UV were noisy for wavelengths < 350 nm, the consistency evident in samples for each meteorological period indicate the features in Fig. 10 reflect ambient aerosols rather than an artifact of the measurement system.  It is also particularly striking that the 2$^{nd}$ UV feature in the Stagnant samples exhibits a minimum at ~ 360 nm (outside of the noisy region of the deuterium lamp) where one would expect a signal from brown carbon consistent with the discussion in Section 3.2.  All of the spectra were blank corrected and only those above the lower limit of detection (LLOD = the mean blank spectrum + 3 standard deviations of the mean) were analyzed.  It may be possible that spectral noise could manifest in a $\sigma_{abs}$ (and hence, $\omega$) spectrum if the values are near the LLOD, however, the appearance of similar features spanning a relatively broad range in wavelengths (compared to the noisy 0.2 nm resolution measurement for $\lambda$ < 350 nm) and appearing in both the UV and visible regions in multiple samples of a given meteorological period suggests the observed features were not attributable to noise.  Future studies where composition measurements that encompass the total aerosol and not just the water-soluble components are anticipated to enable the identification of specific molecular structures and/or compounds responsible for the kinds of features revealed in Fig. 10."*

Note, we have updated the gridlines in the panels of Fig. 10 to clearly delineate 350 nm (i.e., the wavelength of the lamp/noise change) in each example.

*Line 213: Please elaborate the "error propagation".*

Both referees requested elaboration on this statement.  We have added the parenthetical at line 213 (251) to clarify that the uncertainty in each term of Eq. (1) is used in the error estimate for $\sigma_{abs}$.  Further, we now list the uncertainty related to each term in Eq. (1) and add a sentence to state the estimated error in $\sigma_{abs}$ and point to Table 1:

*"Error propagation (i.e., the propagation of the uncertainty of each term in (Eq. 1)) was used to estimate the uncertainty in the $\sigma_{abs}$ spectra (Eq. (1)) from the estimated 15% error of the absorbance spectra, the standard deviation of the slope ($\pm 0.034$) used to estimate $\beta_{K\text{-}OC}$, the uncertainty in the filter area (based on $\pm$ 1 mm in diameter of the estimated filter sampling diameter of 42.6 mm of the  47 mm diameter filters) and the standard deviation of the mean volume of air sampled for each filter calculated from 1 s flow rate data.  Over 300-700 nm the estimated error in $\sigma_{abs}$ is ~17% (Table 1)."*

*Lines 315-316: Do you have any explanations of 'smoothly varying" and "spectral feature" events?*

As we discuss later in the manuscript using the water-soluble composition data, it is likely related to differing organic composition during the different meteorological periods.  However, we need to be careful here as we do not have the kind of aerosol composition data that we would need to characterize water insoluble aerosol components contributing to the total absorption spectra in Fig. 2.  So, we do not have an explanation, but we can hypothesize.  We have added a sentence here (lines 360-362 of the revised manuscript) to offer that hypothesis:

*"The spectral features (i.e., enhanced absorption over a limited wavelength range compared to the rest of the spectrum) likely arise from specific molecular structures within the ambient aerosols that absorb light over a limited wavelength range as discussed in the introduction."*

Also note, in addressing this comment we found an error in the text that was missed at submission.  In two places in Section 3.1 "Fig." should have preceded the number 2, we have updated the text accordingly.  A similar problem occurred in two places where Fig. and Figure were missing preceding the number 3 in Section 3.2.  Those errors have been corrected as well."

*Technical Corrections:*

*Reference Section: The part-1 paper (Jordan et al., 2020b) supposed to be submitted to AMT.*

Thank you for catching this!  We have updated the references in both Parts 1 and 2 with the current citations provided by AMT.  Also, we have provided an update to the Crawford et al., 2020 reference.

**Anonymous Referee #2**

*General Comments:*

*The manuscript provides and discusses the retrievals of aerosol absorption and SSA over a wide wavelength range of 300-700 nm obtained from in situ sampling methods employed during the*

*May-June 2016 Korea United States - Ocean Color (KORUS-OC) cruise conducted in concert with the broader air quality campaign (KORUS-AQ). Various filter-based measurement techniques are applied to obtain total aerosol absorption and soluble-absorption coefficients. Authors further used the extinction retrievals from the Part-I of this study and combined it with the absorption retrievals calculated here to obtain the SSA values. The measurements/retrievals were further put in context of the expected aerosol composition for the different meteorological regimes observed over the period of the campaign. The findings of this study are valuable to the aerosol community because absorption retrievals in the near-UV range are sparse in general and recent studies have highlighted the importance of quantifying the aerosol absorption in the near- UV to visible wavelengths, wherein presence of brown carbon/organic carbon show a steep spectral dependency. However, this study particularly lacks in detail explaining of the absorption features in this important near-UV wavelength range, either due to the noise in the measurement spectra itself or due to the way correction factors for higher wavelength range are extrapolated for these smaller wavelengths. This issue can be addressed by providing clarification statements wherever required, added speculations for observed features in near-UV range and better quantifying the errors associated with the total absorption retrievals in the near-UV wavelength range. I recommend the publication of this manuscript after following suggestions/comments are incorporated.*

As with Referee #1, we thank you for your insights that have greatly improved Section 3.4 and the manuscript overall. We have added more discussion regarding the β correction as we explain in detail below. That discussion should alleviate your concerns about its application over the entire measured wavelength range.

*Specific/major comments:*

*Section 2.1: It is okay to reference Peterson et al. 2019 here for the classification of meteorological regimes that are used in the subsequent analysis, however, a brief description of what each regime means within this section would be more reader-friendly (instead of waiting for this clarification in section 3.2. later). For example, 'Blocking' term/regime is not self-explanatory. This hinders the understanding of Fig. 1 as well (for a reader) that is associated with the text in this section.*

This is an excellent observation. We have added brief details parenthetically in Section 2.1 to orient the reader to these differing meteorological regimes.

*"The locations shown in Fig. 1 are color coded according to the meteorological regimes during the cruise as described by Peterson et al. (2019): Stagnant (May 17th - 22nd, under a persistent anticyclone), Transport/Haze (May 25th-31st, dynamic meteorology with 4 frontal passages, low-level transport from upwind sources in China, and humid conditions promoting the development of fog and haze under cloudy skies), and Blocking (June 1st - 7th, less persistent stagnation arising from a Rex Block, i.e., adjacent high and low pressure systems with the high poleward of the low)..."*

*1. Line 146: "These data were retained for the assessment of the correction factor for the spectral absorption measurement discussed in Sect. 2.3". Can the authors discuss the rationale*

*behind this choice? What implications or bias do they anticipate of including the contaminated measurements in their subsequent assessment of correction factor?*

The rationale to exclude these data from the majority of analyses in the text is to focus on ambient aerosol characteristics rather than the characteristics of our own ship exhaust. However, such emissions do not produce aerosol fundamentally different from other anthropogenic combustion sources. For the filter samples here, the differences in scattering and absorption for the contaminated filters vs. the rest are small (compare the solid and open green symbols in Fig. S1 and bear in mind that the plume interceptions (the gray plus symbols) were generally brief compared to the interval of the filter sample) hence, retention of these filters is intended to provide better statistics and does not introduce bias in the correction factor. However, there was one filter (#22) we excluded from the correction factor calculation due to it being overloaded under the highly polluted ambient conditions that would have introduced a bias had we kept it in the set for that calculation. We have updated the text (starting at line 161 in the revised manuscript )as follows:

*"The degree of contamination depended on the duration of the interception as well as on the aerosol property of interest (e.g., Fig. S1), such that in the few instances where the plume contamination (PF1, gray plus symbols) values in scattering or absorption were relatively large their influence over the filter sampling interval was small (compare the solid, All data, and open, PF0, green symbols in Fig. S1). Here, single scattering albedo calculated from IN101 and TAP data provided the most conservative delineation between ambient and ship plume aerosol (Figs. S1 and S2) resulting in 13 of the filter pairs being rejected (Table S2) from most of the analyses presented in Sect. 3 (except as noted). Given the generally small difference in absorption resulting from the ship plume interceptions, all but filter 22 were retained for the assessment of the correction factor for the spectral absorption measurement discussed in Sect. 2.3. Although, filter 22 was rejected from further analyses on the basis of ship plume contamination (Table S2), this particular filter was also so heavily loaded due to polluted ambient conditions that the measured spectrum in the integrating sphere was distorted. Hence, this filter was excluded from the calculation of the correction factor on the basis of its being overloaded."*

*2. Section 2.3: How accurate or reliable is the use of the correction factor, β, based on σabs/Abs at three specific wavelengths (467, 528, and 652) of TAP in calculating the σabs outside of this wavelength range, specially at UV and near-UV wavelength ranges that are much smaller than 467 nm? Also, suggest moving Figure S3 to being part of the main text, and adding the final βK-OC that was obtained after scaling.*

The correction factors reported in the literature cited in our text are based on a diverse set of aquatic samples with spectral analyses performed over the full wavelength range of the measurement, typically about 300 - 800 nm. It is important to note that the correction is a function of absorbance not wavelength. As we noted in our response to Referee #1's comment about lines 175-177, the ocean color community is also interested in looking further into the question of whether a wavelength dependence ought to be considered to better correct for enhanced scattering in the UV, but at present they do not. Given the differences in particle populations (both in terms of size and composition) in the ambient atmosphere from those in aquatic environments, we think a more thorough evaluation of the appropriate β correction for

atmospheric samples is warranted, but it beyond the scope of this work. For this reason, we prefer to keep Fig. S3 in the supplement rather than the main text as the determination of $\beta_{K\text{-}OC}$ was not a primary objective of this study. Given the differences between the uncorrected and corrected values compared to the TAP measurements shown in Fig. S3 an adjustment to the correction was clearly necessary, but our approach was a simple empirical fit between two measurement techniques without any attempt to identify the specific attributes between the aerosol populations that cause the shift. As we note in the text, it is likely related to the different size ranges of the particles, but that is only conjecture. Moving this figure to the main text would imply a more thorough evaluation of the correction factor than we were able to perform with the data available. We respectfully decline to move the figure, but we have added the final $\beta_{K\text{-}OC}$ value in the text at lines 249-250 of the revised manuscript as follows:

*"However, taking advantage of the TAP data set, $\beta$ for the KORUS-OC data ($\beta_{K\text{-}OC}$) was obtained by scaling $\beta_s$ to fit the TAP data ($\beta_{K\text{-}OC} = 0.593 \times \beta_s = 1.836 \times Abs^{-0.0867}$, see Fig. S3) resulting in quantitative $\sigma_{abs}$ spectra."*

*3. Based on the above point, what impact will the uncertainty in estimation of $\beta$ for near-UV wavelengths, combined with the noisy absorbance measurements in the same wavelength range, have on the overall quality of retrievals for $\sigma_{abs}$ using this filter-retrieval method? As understood by the aerosol community and the authors also point out elsewhere in the manuscript, retrievals in this wavelength-range are particularly important for the estimation of absorption by brown carbon/organic carbon aerosol particles. The manuscript needs more elaborate discussions on this aspect.*

As stated in our response to the above point, the $\beta$ correction is a function of absorbance not wavelength. Regarding the concern about the UV noise, please see our response to Referee #1's comment on lines 175-177 of the submitted manuscript. We agree, that the UV part of the spectrum is what is of most interest here to the atmospheric community given the lack of such data for ambient aerosols. We are continuing to pursue this line of inquiry and anticipate the acquisition of data from diverse ambient atmospheric environments will allow for a more expansive discussion of this topic in future work.

*4. Line 213-14: This sentence on error propagation is hard to understand from the way it is currently formulated. From equation 1, apart from the uncertainty in absorbance, there is also an uncertainty associated with scaling of $\beta$ and may be further in the extrapolation of $\beta$ for smaller wavelengths (if at all accounted for).*

Thank you. Both you and Referee #1 pointed out that we did not fully describe this calculation as we should have done. We have revised the text starting at line 213 (251) to as follows:

*"Error propagation (i.e., the propagation of the uncertainty of each term in (Eq. 1)) was used to estimate the uncertainty in the $\sigma_{abs}$ spectra (Eq. (1)) from the estimated 15% error of the absorbance spectra, the standard deviation of the slope ($\pm 0.034$) used to estimate $\beta_{K\text{-}OC}$, the uncertainty in the filter area (based on $\pm$ 1 mm in diameter of the estimated filter sampling diameter of 42.6 mm of the  47 mm diameter filters) and the standard deviation of the mean*

*volume of air sampled for each filter calculated from 1 s flow rate data. Over 300-700 nm the estimated error in $\sigma_{abs}$ is ~17% (Table 1)."*

*5. Section 3.1: I wonder if this section should be discussed in the light of discussions that follow regarding the differences in aerosol composition and size between the different meteorological regimes for better understanding of the differences between the absorption spectra obtained from the filters using different methods. For example, it is hard for the readers to move forward with the reading while questions about, why $\sigma$MeOH-abs is above detection, while little of the $\sigma$DI-abs is above detection for the Transport/Haze example lingers on. It would be nice to either include the speculated reasons for these differences within this section and explain the reasons for these speculations further in the following sections or rather discuss all of the absorption spectra differences in the light of what we know about the composition and size distribution of aerosol particles in each meteorological regime. Some of the differences in Figure2, mentioned in section 3.1 are not even addressed later on.*

This is an excellent suggestion. Referee #1 also requested a bit more discussion of the total absorption features (see our response to their line 315-316 comment). However, your comment motivated the replacement of the sentence that had ended Section 3.1 in the submitted manuscript with two new sentences (starting at line 368 in the revised manuscript) to address the differences in the soluble spectra, along with a new paragraph to lay out what's to come in the remainder of Section 3 as follows:

*"The differences between the spectra obtained from the DI and MeOH extracts for any given sample arises from differing solubilities of the chemical components of the aerosols in those two solvents. Differences in the soluble spectra across the sample set arise from differences in the ambient aerosol population across the meteorological regimes of the campaign.*

*DI extracts were analyzed for composition enabling analyses to relate composition to optical properties as will be discussed in Section 3.2. Also in that section, previously published work from the KORUS-AQ campaign is used to provide greater context for the differences in the aerosol populations sampled during the three meteorological regimes that occurred during the cruise. In Section 3.3 fits to the spectra set from 2nd order polynomials are compared to those from power laws. This section also includes a preliminary examination of how the different aerosol populations map into the two parameter space described by the 2nd order polynomial coefficients providing a potential avenue to relate aerosol physicochemical properties to spectral optical characteristics. In Section 3.4, single scattering albedo ($\omega$) spectra calculated from $\sigma_{abs}$ and $\sigma_{ext}$ are used to illustrate the spectral features in $\sigma_{abs}$ that more clearly manifest in $\omega$. The in situ aerosol instrument suite aboard the R/V Onnuri did not include on-line aerosol composition measurements that could have been used to interpret the absorption features in the $\sigma_{abs}$ spectra limiting chemical attribution to the various features observed. However, the examples provided show that future studies that combine hyperspectral total absorption with composition measurements that capture the total aerosol (not just the water-soluble components) may enable such linkages to be made."*

*6. Section 3.4: Figure 10 and this section is possibly one of the most useful sections for aerosol community as measurements/retrievals of SSA ($\omega$) across a wide range of wavelengths (300- 700*

*nm) can provide a significant constraint on aerosol modeling efforts. Suggest highlighting this part in the abstract for larger appeal. Also, spectral variation in SSA in near-UV range for multiple instances/examples in Figure 10 is interesting. How much of this variation is attributed to the uncertainty in retrieval methods for this particular wavelength range and how much of this is a real feature? This should be discussed in greater detail, rather than leaving this part just as an observation and proposing some curve-fitting exercise.*

Referee #1 had the same valuable insight and we concur with you both that we gave Fig. 10 short shrift both in the Abstract and Section 3.4.  Please see our response to Referee #1's comments about the Abstract lines 36-37 and lines 175-177 of the submitted manuscript where we include revised text both for the Abstract and a paragraph to address the noise question (now added to Section 3.4).  In addition, we added a new final sentence to Section 3.4 at line 601 of the revised manuscript because the purpose of Fig. 10 is show that mathematical functional representations are insufficient and that measured spectra are needed for $\sigma_{abs}$ and $\omega$:

*"However, whatever mathematical approaches may prove useful, what Fig. 10 primarily reveals is the need for hyperspectral measurements to capture the variability present in $\sigma_{abs}$ and $\omega$ spectra of ambient aerosol populations.  Neither power laws nor 2$^{nd}$ order polynomials can reproduce the features observed in this data set."*

*Minor/Editorial comments:*

*Line 109: Is 'package' the best suitable word to describe a suite of instruments? 'Instrument suite' or something similar sounds more appropriate.*

Changed "measurement package" to "instrument suite".  Note, we made this change in a few of other places as well (see the tracked changes in the revised manuscript).  The only place "package" is still found is in reference to a package of Teflon filters.

*Line 27: It is not specified previous to this sentence what DI stands for.*

DI (deionized water) and MeOH (methanol) are defined in the sentence starting on line 23 "Part 2 expands on...") so we have not defined it again in the sentence on line 27.

*Line 46: "......aerosol size distributions than **cannot** be obtained from single Ångström exponents alone.*

The suggested revision appears to be a misreading of the sentence.  Either one can say "than can be obtained" or "that cannot be obtained", but to say "than cannot be obtained" is not correct.  We have kept the original text.

*Line 51: than Ångström exponents (AE)-> compared to the AE approach*

We replaced "than" at line 51 (60) with " compared to that from".

*Line 54: This sentence is confusing. Suggested change: The data reported in this study were from in situ aerosol measurements in a package deployed aboard the ...-> The data reported in this study are in situ aerosol measurements from a suite of instruments deployed aboard the....*

Agreed, thanks!  The sentence at line 54 (62) has been updated as recommended and a reference to the new Table 1 has been added:

*"The data reported in this study are in situ aerosol measurements (Table 1) from a suite of instruments deployed aboard the R/V Onnuri during the Korea United States - Ocean Color (KORUS-OC (US-Korean Steering Group, 2015)) cruise around the Korean peninsula (May 20th to June 6th in 2016)."*

*Line111: ..... influenced the observed spectral characteristics observed aboard the R/V Onnuri. -> influenced the observed spectral characteristics obtained from the instruments aboard the R/V Onnuri.*

Changed line 111 (122-123) as recommended.

*Line 121: Insert a Figure 1 reference right here.*

Added at the end of the sentence that starts on line 121 (132):

*"In brief, the KORUS-OC cruise of the R/V Onnuri sailed first along the east coast of South Korea, then transited to the west (Fig. 1)."*

*Line 122: ....50% size cut of 1.3 μm diameter. -> 50% cut-off size of 1.3 μm diameter.*

Changed as recommended at line 122 (133).

*Line 139: "The high-resolution data set was 'filtered' to remove interceptions of the R/V Onnuri's own ship stack emissions (Jordan et al., 2020b)". I think the use of term 'filtered' here can be confusing. Suggest using 'screened' or another synonym.*

Very good observation, thank you.  We have revised the sentence at line 139 (158) as follows:

*"The high-resolution data set was flagged to identify and remove interceptions of the R/V Onnuri's own ship stack emissions (Jordan et al., 2020b)."*

*Line 164: OC -> Ocean Color (first time occurrence of OC as a noun)*

This is the only place where we use the acronym OC in this way, so we have replaced it by spelling it out instead (now line 192 of the revised manuscript).

*Line 346: Is there a way to identify the downwind and upwind side of the peninsula in Figure 1 (e.g. plotting MERRA-2 (or any other reanalysis) wind vectors over the map). It is hard to*

*understand which part of the plot is being refereed as downwind or upwind in Figure 3 and Figure 4 as well.*

We don't understand this suggestion. The Fig. 1 caption states "...Transport/Haze data split between upwind (west, orange) and downwind (east, yellow) positions around the peninsula..." so the definition of what is considered upwind and downwind of Korea is not only stated in the text, but color-coded in the map. The same color code is used in Fig. 4. Further, in the first paragraph of Section 3.2 (lines 330-331 of the submitted draft) we state, "During the Transport/Haze period transport from the west/northwest brought polluted air from China to S. Korea", which along with Fig. 1 should make clear that upwind is to the west and downwind is to the east. Adding wind vectors to Fig. 1 in our opinion would be more confusing that the current presentation, because the appropriate vectors for Transport/Haze would not apply to samples collected during the other meteorological periods. It would be difficult to show vectors for all 3 periods on Fig. 1 with any clarity. Finally, the ship track by date is shown in Part 1. Here, when Fig. 1 is introduced in the text we state that "...the *R/V Onnuri* sailed first along the east coast of South Korea, then transited to the west (Fig. 1)". In the text you reference (around line 346) we also specify when the ship was downwind and upwind of the peninsula by date consistent with the x-axis in the time series of Fig. 3. For all of these reasons we are puzzled by how the manner in which we distinguish the downwind and upwind subsets of the Transport/Haze period is not clear. We respectfully decline to add wind vectors to Fig. 1. Nonetheless, we have added the following parentheticals at the start of the discussion of Fig. 4 (line 355 (416)) to aid the reader:

[revised manuscript text omitted]

... [1]

*Supplemental Information for*

**New In Situ Aerosol Hyperspectral Optical Measurements over 300-700 nm, Part 2: Extinction, Total Absorption, Water- and Methanol-soluble Absorption observed during the KORUS-OC cruise**

Carolyn E. Jordan[1,2], Ryan M. Stauffer[3], Brian T. Lamb[4], Michael Novak[3,5], Antonio Mannino[3], Ewan C. Crosbie[2,6], Gregory L. Schuster[2], Richard H. Moore[2], Charles H. Hudgins[2], Kenneth L. Thornhill[2,6], Edward L. Winstead[2,6], Bruce E. Anderson[2], Robert F. Martin[2], Michael A. Shook[2], Luke D. Ziemba[2], Andreas J. Beyersdorf[2,7], Claire E. Robinson[2,6], Chelsea A. Corr[2,8], and Maria A. Tzortziou[3,4]

[1]National Institute of Aerospace, Hampton, Virginia, United States of America
[2]NASA Langley Research Center, Hampton, Virginia, United States of America
[3]NASA Goddard Space Flight Center, Greenbelt, Maryland, United States of America
[4]City University of New York, New York, New York, United States of America
[5]Science Systems and Applications Inc., Lanham, Maryland, United States of America
[6]Science Systems and Applications Inc., Hampton, Virginia, United States of America
[7]California State University, San Bernardino, California, United States of America
[8]Springfield College, Springfield, Massachusetts, United States of America

*Correspondence to*: C. E. Jordan (Carolyn.Jordan@nasa.gov)

**Table S1.** Filter number, sampling interval, mean location with standard deviation, and flags for the absorption spectra (-8888 = below detection; -9999 = missing). There were no flags needed for the filter mean SpEx extinction spectra. Gray font indicates filters excluded from analyses due to ship exhaust plume contamination (see below).

| Filter Number | Date & Time (KST) Start | Stop | Latitude (°N) Mean | St. Dev. | Longitude (°E) Mean | St. Dev. | Total Abs. | DI-soluble Abs. | MeOH-soluble Abs. |
|---|---|---|---|---|---|---|---|---|---|
| 1 | 5/22/16 6:22 | 5/22/16 9:08 | 37.0115 | 0.0052 | 130.7049 | 0.0044 | | | |
| 2 | 5/22/16 9:10 | 5/22/16 12:14 | 37.0291 | 0.0052 | 130.7205 | 0.0107 | | | |
| 3 | 5/22/16 12:15 | 5/22/16 15:25 | 37.0329 | 0.0043 | 130.7496 | 0.0066 | | | |
| 4 | 5/22/16 15:27 | 5/22/16 18:21 | 37.0241 | 0.0071 | 130.7712 | 0.0163 | | | |
| 5 | 5/22/16 18:23 | 5/23/16 6:35 | 36.9782 | 0.0213 | 130.8028 | 0.0192 | | | |
| 6 | 5/23/16 6:39 | 5/23/16 9:21 | 37.2664 | 0.1088 | 131.1076 | 0.1130 | | | -9999 |
| 7 | 5/23/16 9:23 | 5/23/16 12:14 | 37.5483 | 0.0452 | 131.3533 | 0.0298 | | | -9999 |
| 8 | 5/23/16 12:15 | 5/23/16 15:21 | 37.7078 | 0.0534 | 131.2631 | 0.0322 | | | -9999 |
| 9 | 5/23/16 15:22 | 5/23/16 17:55 | 37.8527 | 0.0391 | 131.1205 | 0.0399 | | | -9999 |
| 10 | 5/24/16 6:26 | 5/24/16 9:19 | 37.0772 | 0.0103 | 131.0734 | 0.0906 | | | -9999 |
| 11 | 5/24/16 9:21 | 5/24/16 12:10 | 37.1090 | 0.0117 | 130.8724 | 0.0820 | | | |
| 12 | 5/24/16 12:12 | 5/24/16 15:13 | 37.1455 | 0.0095 | 130.6959 | 0.0826 | | | -9999 |
| 13 | 5/24/16 15:14 | 5/24/16 18:16 | 37.1934 | 0.0376 | 130.4023 | 0.1285 | | | |
| 14 | 5/24/16 18:18 | 5/25/16 6:20 | 37.5795 | 0.1009 | 129.9108 | 0.1136 | | | -9999 |
| 15 | 5/25/16 6:23 | 5/25/16 9:12 | 37.6988 | 0.0126 | 129.7585 | 0.0434 | | | |
| 16 | 5/25/16 9:13 | 5/25/16 12:01 | 37.6856 | 0.0086 | 129.6430 | 0.0125 | | | -9999 |
| 17 | 5/25/16 12:02 | 5/25/16 15:01 | 37.7793 | 0.0332 | 129.6209 | 0.0160 | | | -9999 |
| 18 | 5/25/16 15:03 | 5/25/16 17:54 | 37.8287 | 0.0090 | 129.5813 | 0.0168 | | | -9999 |
| 19 | 5/25/16 17:55 | 5/26/16 6:20 | 37.5098 | 0.4201 | 129.7468 | 0.2728 | | | -9999 |
| 20 | 5/26/16 6:23 | 5/26/16 9:24 | 36.5316 | 0.0061 | 130.2587 | 0.0322 | | | |
| 21 | 5/26/16 9:25 | 5/26/16 12:13 | 36.4911 | 0.0376 | 130.1324 | 0.0464 | | | |
| 22 | 5/26/16 12:14 | 5/26/16 15:08 | 36.4278 | 0.0289 | 130.0288 | 0.0588 | | | |
| 23 | 5/28/16 9:33 | 5/28/16 12:26 | 34.0779 | 0.0494 | 128.1513 | 0.1000 | | | -9999 |
| 24 | 5/29/16 6:49 | 5/29/16 9:27 | 35.0119 | 0.0107 | 125.0027 | 0.0030 | | | -9999 |
| 25 | 5/29/16 9:29 | 5/29/16 12:22 | 35.0017 | 0.0060 | 124.9125 | 0.0685 | | | |
| 26 | 5/29/16 12:23 | 5/29/16 15:30 | 34.9889 | 0.0101 | 124.7036 | 0.0638 | | | |
| 27 | 5/29/16 15:31 | 5/29/16 18:31 | 35.1041 | 0.0299 | 124.5085 | 0.0545 | | | |
| 28 | 5/29/16 18:32 | 5/30/16 6:21 | 35.0589 | 0.0163 | 124.3375 | 0.0103 | | | |
| 29 | 5/30/16 6:23 | 5/30/16 9:08 | 35.0473 | 0.0238 | 124.3578 | 0.0111 | -8888 | | |
| 30 | 5/30/16 9:09 | 5/30/16 12:11 | 35.0244 | 0.0347 | 124.3456 | 0.0163 | | | |
| 31 | 5/30/16 12:12 | 5/30/16 15:08 | 35.0684 | 0.0072 | 124.3714 | 0.0041 | | | -9999 |
| 32 | 5/30/16 15:09 | 5/30/16 18:08 | 35.0331 | 0.0106 | 124.3750 | 0.0081 | | | -9999 |
| 33 | 5/30/16 18:09 | 5/31/16 6:16 | 35.2348 | 0.1810 | 124.5618 | 0.1827 | | | -9999 |
| 34 | 5/31/16 6:17 | 5/31/16 9:07 | 35.6493 | 0.0300 | 124.9787 | 0.0295 | | -8888 | |
| 35 | 5/31/16 9:08 | 5/31/16 12:07 | 35.6675 | 0.0027 | 125.1055 | 0.0682 | | -8888 | |

| 36 | 5/31/16 12:08 | 5/31/16 15:08 | 35.6621 | 0.0037 | 125.3913 | 0.1176 | | -8888 | -9999 |
|----|---------------|---------------|---------|--------|----------|--------|---|-------|-------|
| 37 | 5/31/16 18:24 | 6/1/16 6:20 | 36.3116 | 0.3670 | 125.4494 | 0.2323 | | | |
| 38 | 6/1/16 6:21 | 6/1/16 9:06 | 36.8974 | 0.0480 | 125.2866 | 0.0324 | | | -9999 |
| 39 | 6/1/16 9:06 | 6/1/16 12:13 | 37.2150 | 0.1145 | 125.5746 | 0.0948 | | | |
| 40 | 6/1/16 12:13 | 6/1/16 15:08 | 37.3512 | 0.0042 | 125.6766 | 0.0101 | | | -9999 |
| 41 | 6/1/16 15:09 | 6/1/16 18:00 | 37.3373 | 0.0065 | 125.6571 | 0.0138 | | | |
| 42 | 6/1/16 18:01 | 6/2/16 6:11 | 36.5614 | 0.5072 | 125.0593 | 0.3295 | | | |
| 43 | 6/2/16 6:13 | 6/2/16 9:09 | 35.4287 | 0.1008 | 124.3942 | 0.0681 | | | |
| 44 | 6/2/16 9:10 | 6/2/16 12:06 | 35.2375 | 0.0667 | 124.3276 | 0.0070 | | | |
| 45 | 6/2/16 12:07 | 6/2/16 15:10 | 35.0508 | 0.0669 | 124.3396 | 0.0044 | | | -9999 |
| 46 | 6/2/16 15:11 | 6/2/16 18:04 | 34.8459 | 0.0738 | 124.3356 | 0.0083 | | | |
| 47 | 6/2/16 18:05 | 6/3/16 6:13 | 33.9574 | 0.4177 | 124.2784 | 0.0272 | | | |
| 48 | 6/3/16 6:15 | 6/3/16 9:08 | 33.4690 | 0.0084 | 124.2439 | 0.0123 | | | |
| 49 | 6/3/16 9:09 | 6/3/16 12:09 | 33.4798 | 0.0102 | 124.2032 | 0.0083 | | | |
| 50 | 6/3/16 12:10 | 6/3/16 15:07 | 33.5074 | 0.0052 | 124.2043 | 0.0092 | | | -9999 |
| 51 | 6/3/16 15:08 | 6/3/16 17:51 | 33.4949 | 0.0091 | 124.2353 | 0.0079 | | -9999 | -9999 |
| 52 | 6/3/16 17:51 | 6/4/16 6:14 | 33.4971 | 0.0377 | 124.1984 | 0.0370 | | | |
| 53 | 6/4/16 6:16 | 6/4/16 9:19 | 33.5051 | 0.0074 | 124.7797 | 0.2245 | -8888 | -8888 | -9999 |

*Ship plume influence on filter samples*.  In Part 1, ship exhaust plume interceptions were flagged (PF = 1 for plume interception, = 0 for ambient conditions) based on TAP absorption data and gas phase measurements (Jordan et al., 2020b).  For the 53 filter pairs collected, 15 filter sampling intervals did not have any ship plume interceptions at all (PF = 0 throughout the interval).  Little difference was observed in scattering between filter means calculated with and without PF=1 (green symbols for all data and PF = 0, respectively, top panel Fig. S1), due to the fleeting nature of the plume interceptions over the 3- or 12-hour sampling intervals and the relatively minor enhancement in scattering coefficients for PF = 1 periods versus PF = 0.  However, larger differences were found in absorption for 8 filters due to much larger differences between ambient and ship plume aerosol absorption (Fig. S1, middle panel).  Further, calculations of single scattering albedo ($\omega$) showed that an additional 5 filters deviated considerably between the two calculations when scattering was small (Figs. S1 and S2).  Taking a conservative approach, all 13 of these filters (Table S2) have been excluded from the data analyses discussed in Section 3, although they remain in the map shown in Fig. 1 and were used in the determination of the correction factor used to adjust the filter-based spectral absorption correction to match the TAP values as described in Section 2.3.

[Figure]

**Figure S1**. 532 nm $\sigma_{scat}$ (top), $\sigma_{abs}$ (middle), and $\omega$ (bottom) calculated for each filter sampling interval using all (PF = 0 and 1) data or just PF = 0 (solid and open green markers, respectively) or PF = 1 data (black plus symbols, top 2 panels only) from the IN101 nephelometer and TAP (denoted NT when used for $\sigma_{ext}$ or $\omega$ calculations) instruments.

[Figure]

**Figure S2.** NT ω(532nm) calculated using all data over the filter sampling interval vs. using only PF = 0.  All filters are shown (left panel), along with only those retained for further analyses (right panel).

**Table S2**.  Filters excluded from further analyses (except as noted above) based on difference in NT ω calculated using PF = 0 only and PF = 0 and 1.

| Meteorological Periods | Filter Numbers | Ship Contaminated Filters | % Removed |
|---|---|---|---|
| Stagnant | 1-4 | -- | 0 |
| transition | 5-14 | 8, 11 | 20 |
| Transport/Haze - East | 15-22 | 15, 16, 22 | 37.5 |
| Transport/Haze - West | 23-36 | 24, 30 | 14.3 |
| Blocking | 37-53 | 39, 45, 48, 49, 50, 52 | 35.3 |
| Total | 53 | 13 | 24.5 |

[Figure]

**Figure S3.** Comparison of $\sigma_{abs}$ between GF/F and TAP measurements without any correction applied to the GF/F measurements ($\beta_1$, black pluses and line fit) and corrected using $\beta_S$ (red circles and line fit). The 1:1 line (thick black line) is shown for comparison.

**Table S3.** Look up chart to convert LN (wavelengths) to wavelengths as shown in Fig. S4.¶
**LN (Wavelength (μm))** ... [1]

[Figure]

**Figure S4.** Example (FN 21) of linear fits (black lines, left panels) versus 2nd order polynomial fits (black lines, right panels) to measured spectra (red curves, all panels): $\sigma_{ext}$ (top row), $\sigma_{abs}$ (2nd row), $\sigma_{DI-abs}$ (3rd row), and $\sigma_{MeOH-abs}$ (bottom row). The residuals from each fit (blue curves) are shown above each curve fit. Note, FN21 was the example used in Fig. 2 for the Transport/Haze - East case. The x-axis labels of -1.2, -1.0, -0.8, -0.6 and -0.4 for LN($\lambda$($\mu$m)) equal 0.301, 0.368, 0.449, 0.549, 0.670, and 0.698 um wavelengths, respectively.

[Figure]

**Figure S5.** Maps of $\lambda_{ch}$ (left panels) and the $\alpha$ (right panels) in $a_2$ vs. $a_1$ space for the range of values obtained from $\sigma_{abs}$ (top row), $\sigma_{ext}$ ($2^{nd}$ row), $\sigma_{DI-abs}$ ($3^{rd}$ row), and $\sigma_{MeOH-abs}$ (bottom row) spectra. The mapping of $\alpha$ in $a_2$ vs. $a_1$ space rotates as a function of $\lambda_{ch}$ such that a narrow range (e.g. $\lambda_{ch}(\sigma_{abs}) = 0.47$, top left) maps to parallel lines in $\alpha$ (top right), while a wider range of values can map into a fan around $a_2 = 0$ (e.g., $\sigma_{ext}$, $2^{nd}$ row) or distinct branches (e.g., $\sigma_{DI-abs}$ or $\sigma_{MeOH-abs}$, bottom 2 rows) compare the right panels here to the data shown in Fig. 7 of the main text. The color bars used in each of the $\alpha$ maps (right) match those of the Fig. 7 panels.

**Page 6: [1] Deleted**          **Jordan, Carolyn**          11/20/20 12:08:00 PM